EMBO
Molecular Medicine

# A high-throughput RNAi screen for detection of immune-checkpoint molecules that mediate tumor resistance to cytotoxic T lymphocytes

Nisit Khandelwal[1,*,†], Marco Breinig[2,3,†], Tobias Speck[1], Tillmann Michels[1], Christiane Kreutzer[4], Antonio Sorrentino[1], Ashwini Kumar Sharma[5], Ludmila Umansky[1], Heinke Conrad[4], Isabel Poschke[6], Rienk Offringa[6], Rainer König[5,7,8], Helga Bernhard[9], Arthur Machlenkin[10], Michael Boutros[2,3] & Philipp Beckhove[1,**]

## Abstract

The success of T cell-based cancer immunotherapy is limited by tumor's resistance against killing by cytotoxic T lymphocytes (CTLs). Tumor-immune resistance is mediated by cell surface ligands that engage immune-inhibitory receptors on T cells. These ligands represent potent targets for therapeutic inhibition. So far, only few immune-suppressive ligands have been identified. We here describe a rapid high-throughput siRNA-based screening approach that allows a comprehensive identification of ligands on human cancer cells that inhibit CTL-mediated tumor cell killing. We exemplarily demonstrate that CCR9, which is expressed in many cancers, exerts strong immune-regulatory effects on T cell responses in multiple tumors. Unlike PDL1, which inhibits TCR signaling, CCR9 regulates STAT signaling in T cells, resulting in reduced T-helper-1 cytokine secretion and reduced cytotoxic capacity. Moreover, inhibition of CCR9 expression on tumor cells facilitated immunotherapy of human tumors by tumor-specific T cells *in vivo*. Taken together, this method allows a rapid and comprehensive determination of immune-modulatory genes in human tumors which, as an entity, represent the 'immune modulatome' of cancer.

**Keywords** cancer immunotherapy; immune suppression; RNAi screen
**Subject Categories** Biomarkers & Diagnostic Imaging; Cancer; Immunology

## Introduction

Peripheral immune tolerance is important to prevent autoimmune disorders. However, tumor cells use immune checkpoints to prevent immune recognition (Zitvogel *et al*, 2006; Rabinovich *et al*, 2007). Blocking antibodies against surface-expressed immune-regulatory proteins, such as CTLA4 and PD-L1 (Chambers *et al*, 2001; Blank *et al*, 2004), boost anti-tumor immunity and are successfully applied in clinical trials (van Elsas *et al*, 1999; Weber, 2007; Brahmer *et al*, 2012; Topalian *et al*, 2012). Still, treatment unresponsiveness is frequent among patients (Topalian *et al*, 2012), indicating that other immune-checkpoint pathways may be active. Therefore, successful cancer immunotherapy requires a systematic delineation of the entire immune-regulatory circuit—the 'immune modulatome'— expressed on tumors (Woo *et al*, 2012; Berrien-Elliott *et al*, 2013).

A comprehensive detection of immune-checkpoint molecules has been technically challenging due to the lack of robust high-throughput assays that enable a qualitative and quantitative analysis of heterologous interactions between tumor cells and T cells. Screening strategies before have relied on interferon-gamma (IFN-γ)

1 Division of Translational Immunology, German Cancer Research Center (DKFZ), Heidelberg, Germany
2 Division of Signaling and Functional Genomics, German Cancer Research Center (DKFZ), Heidelberg, Germany
3 Department of Cell and Molecular Biology, Faculty of Medicine Mannheim, Heidelberg University, Heidelberg, Germany
4 Division of Immunogenetics, German Cancer Research Center (DKFZ), Heidelberg, Germany
5 Division of Theoretical Bioinformatics, German Cancer Research Center (DKFZ), Heidelberg, Germany
6 Department of Molecular Oncology of Gastrointestinal Tumors, German Cancer Research Center (DKFZ) and Division of Pancreas Carcinoma Research, Surgery Clinic of Heidelberg University, Heidelberg, Germany
7 Integrated Research and Treatment Center, Center for Sepsis Control and Care (CSCC), Jena University Hospital, Jena, Germany
8 Leibniz Institute for Natural Products Research and Infection Biology, Hans-Knöll-Institute, Jena, Germany
9 Department of Hematology/Oncology, Klinikum Darmstadt GmbH, Darmstadt, Germany
10 Sharett Institute of Oncology, Hadassah-Hebrew University Hospital, Jerusalem, Israel
*Corresponding author. Tel: +49 6221 56 5085; Fax: +49 6221 56 5280; E-mail: n.khandelwal@dkfz.de
**Corresponding author. Tel: +49 6221 56 5466; Fax: +49 6221 42 3702; E-mail: p.beckhove@dkfz.de
†These authors contributed equally to this study

    

release as an indicator of anti-tumor NK cell activity (Hill & Martins, 2006; Bellucci *et al*, 2012). However, IFN-γ secretion alone by immune cells does not always correlate with cellular cytotoxicity (Bachmann *et al*, 1999; Slifka *et al*, 1999). We therefore established an assay that measures tumor cell lysis mediated by CTLs in a high-throughput coculture setting. Here, we demonstrate that such an approach is feasible and suitable to identify novel immune-modulatory ligands in breast cancer cells.

## Results

### Design of the high-throughput RNAi screen

We transfected MCF7 breast cancer cells stably (MCF7luc) or transiently with luciferase to exploit reduced luciferase activity for detection of tumor cell lysis in a high-throughput format (Fig 1A). To induce tumor lysis, we used both an antigen-dependent as well as antigen-independent system, whereby we employed, respectively, either survivin tumor antigen-specific T cells or bi-specific antibodies which cross-linked the T cell receptor (TCR) associated molecule CD3 on activated CTLs from healthy donors to the cell surface molecule EpCAM on tumor cells (Strauss *et al*, 1999; Fig 1B). To identify genes that protected tumor cells from CTL-induced killing, MCF7luc cells were transfected in 384-well plates with siRNAs before coculture with CD8$^+$ T cells for 18 h. Our luciferase-based cytotoxicity assay (Luc-CTL assay) also included a viability control per gene knockdown to which no CTLs were added in order to exclude genes with intrinsic impact on cell viability. To identify immune-modulating genes, we calculated the difference in luciferase activity between test wells (containing CTLs) and control wells (without CTLs; Fig 1A). PD-L1 served as a positive control in the screens because it mediates strong immune suppression in breast cancer (Dong *et al*, 2002) and in our assays PD-L1 knockdown increased CTL activity without intrinsic influence on tumor cell viability (Fig 1C). Importantly, the extent of tumor cell killing detected by the Luc-CTL assay was comparable to that obtained with a common test of T cell-mediated cytotoxicity, the $^{51}$chromium-release assay (Brunner *et al*, 1968; Fig 1D).

To translate the Luc-CTL assay to a high-throughput RNAi screening approach, we focused on a library of 520 genes coding for transmembrane and cell surface proteins as these are suitable targets for therapeutic function-blocking antibodies. The candidate identification procedure is outlined in Fig 2A. We conducted the screen not only with MCF7luc cells (screen 1) but also with transiently Luc-transfected cells (screen 2) in parallel, as the latter approach can be easily employed for screening of various tumor cell lines. Moreover, given the limitations in expanding antigen-specific T cells, we also tested the feasibility of using polyclonal, pre-activated CTLs from healthy donors as effector T cell population with the aid of bi-specific antibody. Using the latter approach, we could not only identify tumor-associated immune suppressors that suppress T cell function, but also targets that could further aid in better clinical success of the anti-EpCAM × CD3 bi-specific antibody. Each screen contained a set of 4 replicates, two of which were exposed to CTLs (toxicity set) and two were incubated without CTLs (viability set). Finally, we employed data from an additional screen (based on CTG assay) in which cell viability was determined

independent of luciferase activity by measuring intracellular ATP levels (screen 4) and genes that impacted cell viability in this assay were also excluded from the final hit list (Supplementary Fig S1A).

The reproducibility between the replicates within the individual screens was high for both the toxicity set and the viability set and is represented for screen 2 in Fig 2B (Pearson rank correlation coefficient: 0.73 and 0.94, respectively). An overview of results for each gene from the individual screens is depicted in Fig 2C–E. Luciferase-targeting siRNA (FLuc) expectedly abrogated the luciferase signal under both conditions and served as an internal control for the luciferase-based readout. As anticipated, control siRNAs targeting genes indispensable for cell survival (UBC, PLK-1) induced prominent cell killing (without CTLs; *x*-axis; Fig 2C–E). In contrast, negative control siRNAs did not affect the luciferase signal. Thus, UBC and PLK-1 on one hand and negative control siRNA1 and siRNA2 on the other hand defined a range among which candidate genes were ranked according to their impact on CTL-mediated tumor lysis. Silencing of genes with reported immune-regulatory function, namely PD-L1, CEACAM-6, and galectin-3 (GAL-3), strongly reduced luciferase activity only in the cytotoxicity setup, whereby PD-L1 showed a higher impact on tumor lysis (Blank *et al*, 2004; Peng *et al*, 2008; Witzens-Harig *et al*, 2013). These therefore served as immune-modulatory reference genes throughout the screens. Next, unsupervised hierarchical clustering of the differential score between toxicity and viability values for all genes across the different screens was used to identify candidate immune modulators (Fig 3). Clustering analysis revealed heterogeneity in the immune-modulatory properties of certain genes when compared across the 3 screens, which was expected given the intentional assorted biological setup used for the 3 different screens, including CTL source and tumor cell modification for luciferase expression. Therefore, only those robust immune-regulatory genes that modified anti-tumor-immune response irrespective of the T cell source or tumor modification were shortlisted for further filtering based on CTG assay to exclude hits that revealed viability effects. Rigorous filtering resulted in the identification of high-confidence hits of immune suppressors as well as immune activators associated with breast cancer.

### Validation of immune-modulatory function of CCR9

For exemplary functional validation of the screening approach, we chose C-C chemokine receptor type 9 (CCR9) as it was found to be highly immunosuppressive in all the three screens despite the divergent biological background, inhibiting T cell function in an antigen-dependent as well as antigen-independent manner (Fig 3). CCR9 is a chemokine receptor involved in immune cell trafficking (Kunkel *et al*, 2000; Uehara *et al*, 2002) and is expressed on tolerogenic plasmacytoid dendritic cells (Hadeiba *et al*, 2008). So far, an implication of CCR9 in T cell function or tumor-immune resistance has not been reported.

The mRNA and protein knockdown efficiency of single siRNAs within the CCR9 siRNA pool correlated well with the functional effect on T cell cytotoxicity (Fig 4A and B), while none of the CCR9 siRNAs influenced cell viability (Supplementary Fig S1B). Surface expression of CCR9 on MCF7 cells was also found to be reduced by 50% in flow cytometry staining using CCR9 s1 siRNA. Knockdown of CCR9 using siRNA markedly increased MCF7 lysis by survivin-specific CTL (Fig 4C) in the classical chromium-release

    

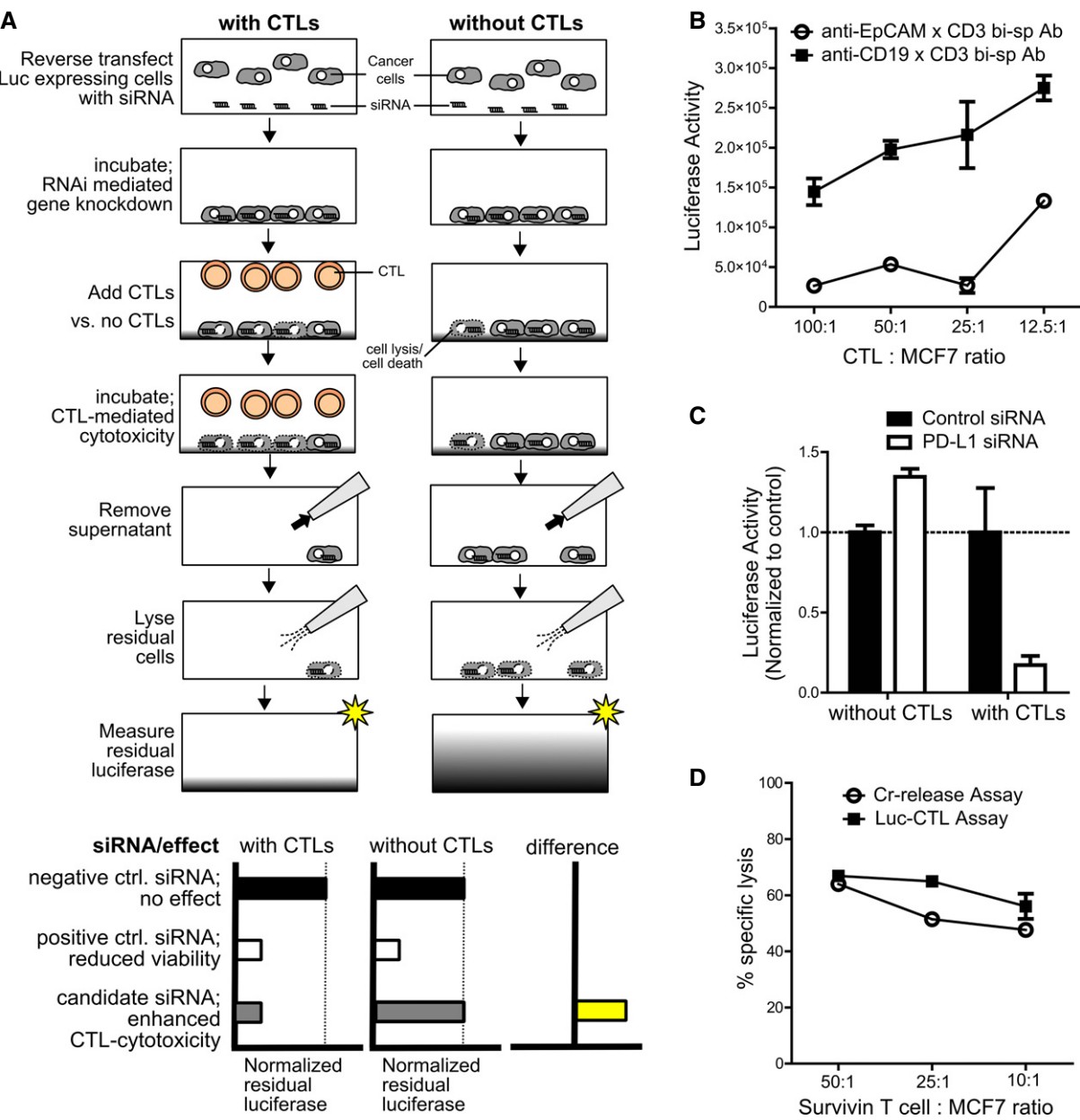

**Figure 1.  Luc-CTL assay design used for identification of immune-checkpoint molecules.**

A  Assay principle: RNAi was performed with luciferase-expressing cells that were challenged with or without CTLs and bi-specific antibody. Before readout, cell supernatant was removed and the remaining intact cells were lysed to measure the residual-cell-associated luciferase. To identify immune-checkpoint regulators, the difference between normalized luciferase measurements for conditions with CTLs and without CTLs was calculated. siRNAs enhancing CTL cytotoxicity would only reduce normalized luciferase levels under conditions with CTLs; hence, the difference between luciferase measurements will be > 0.

B  Luc-CTL assay performed at different T cell to MCF7 cell ratio with PBMC-derived CD8[+] T cells and anti-EpCAM x CD3 bi-specific antibody (○). Anti-CD19 × CD3 bi-specific Ab (■) was used as a specificity control since CD19 is a B-lymphocyte-specific antigen and therefore this bsAb fails to cross-link tumor to T cells. Lower luciferase intensity indicates higher lysis. Error bars denote ± SEM.

C  Luc-CTL assay was performed with MCF7 cells transfected with control or PD-L1-specific siRNAs and cocultured with or without CTLs and bsAb. For each condition, the luciferase activity of PD-L1-siRNA-treated cells was normalized to that of the control siRNA; $n = 8$. Error bars denote ± SEM.

D  Comparison between the Luc-CTL assay (■) and the classical chromium (Cr)-release assay (○) with MCF7 as target cells and survivin-specific T cells as effector cells at varying effector to target (E:T) ratios. Error bars denote ± SEM.

assay. Conversely, overexpression of CCR9 inhibited tumor lysis, demonstrating that CCR9 expression enables immune escape of cancer cells (Fig 4D). CCR9 inhibition in MDA-MB-231 metastatic breast cancer cell line also resulted in marked increase in immune-mediated tumor lysis (Fig 4E). To explore the broad applicability of CCR9-mediated immune suppression in different tumor entities under clinical setting, we next silenced CCR9 in patient-derived primary melanoma cells (M579-A2 cells) and cocultured

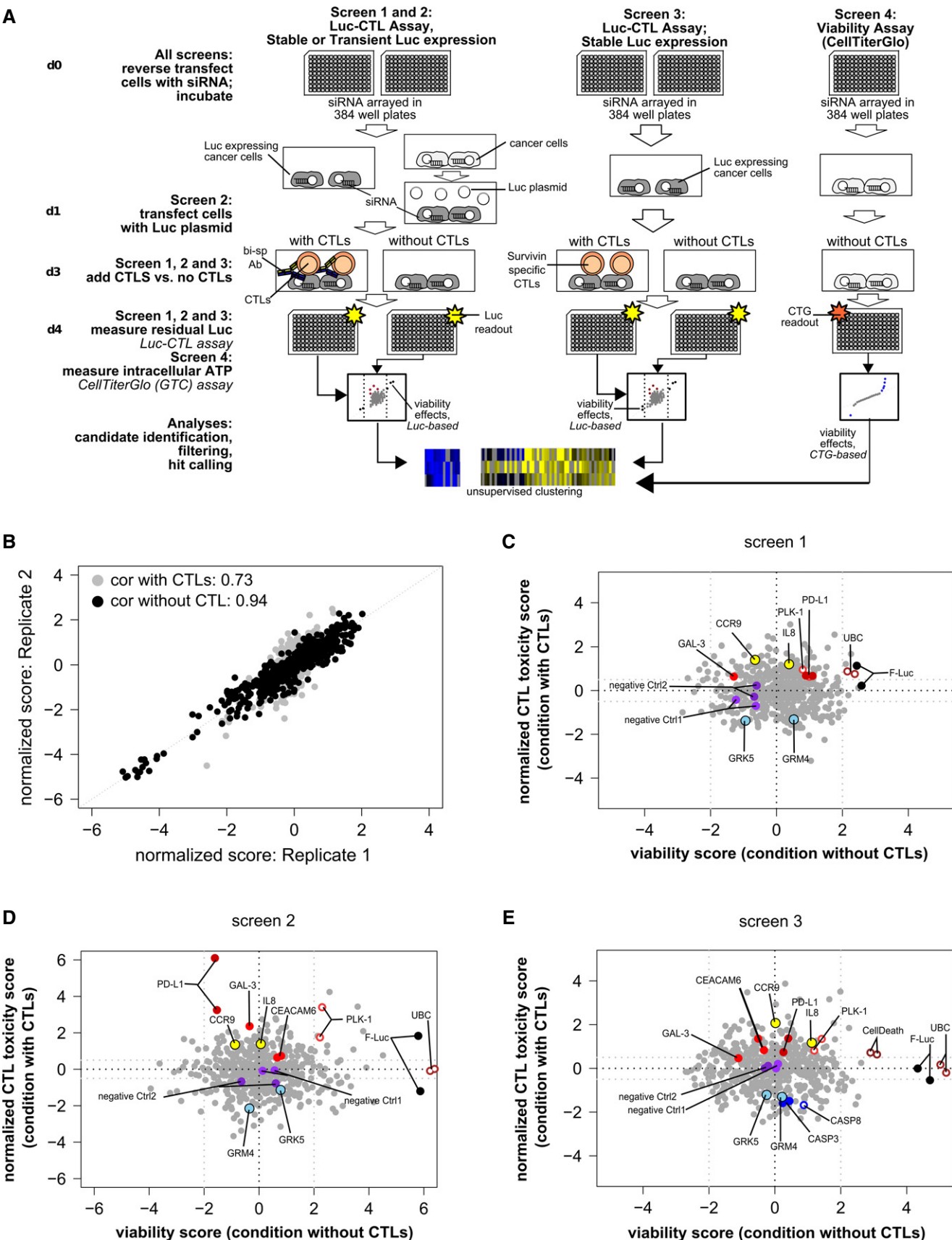

**Figure 2.**

**Figure 2.   Layout and analysis of the RNAi screen used to identify immune-modulatory tumor genes.**

A    Workflow and experimental layout of the RNAi screen which was performed thrice, each time in duplicates, along with an additional CTG-based viability screen to filter out lethal genes from the final hit list. Hits were analyzed after data normalization using the cellHTS2 package.

B    Correlation between replicates for screen 2 for both viability and toxicity sets, as determined by the Pearson rank correlation test.

C–E   Graphical summary of gene function related to modification of T cell-mediated tumor lysis and cell viability for screens 1, 2, and 3, respectively. Positive score = reduced cancer cell viability; negative score = increased viability. X-axis: influence on cell viability without addition of T cells. Y-axis: influence on cell viability with addition of T cells. Appropriate immune-suppressive (PD-L1, CEACAM-6, GAL-3) and lethality (UBC, PLK-1) controls, along with few positive (GRM4, GRK5) and negative (CCR9, IL8) candidate immune-modulatory hits are highlighted herein.

them with HLA-matched tumor-infiltrating lymphocytes (TIL; clone 412) derived from melanoma patient and found a remarkable increase in melanoma cell lysis upon CCR9 knockdown in comparison to the control knockdown (Fig 4F). Similarly, HLA-matched TIL cultures (TIL 53) from pancreatic adenocarcinoma patients recognized and lysed PANC-1 pancreatic cancer cells more effectively upon CCR9 knockdown as shown in Fig 4G, stressing that CCR9-mediated immune suppression may be a clinically relevant phenomenon in multiple tumor entities.

We finally explored the influence of CCR9 expression on CTL functions. CCR9 knockdown in MCF7 cells significantly increased the secretion of IFN-γ and granzyme B by survivin-specific CTL in response to MCF7 cells (Fig 5A and B), supporting the increased cytotoxicity observed in the kill assays. To assess whether this

correlated with increased TCR activation and signaling, we performed TCR phospho-plex analysis in survivin-specific CTLs after contact with CCR9$^{hi}$ or CCR9$^{lo}$ MCF7 cells. With the exception of some degree of reduced Lck phosphorylation (which was detectable only 5 min after exposure to CCR9$^{lo}$ tumor cells), we did not observe any CCR9-dependent changes in TCR signaling (Supplementary Fig S2A and B). Nevertheless, TCR engagement was found to be necessary for CCR9-mediated immunosuppression as polyclonal T cells failed to secrete higher levels of IFN-γ in response to CCR9$^{lo}$ MCF7 cells in the absence of anti-EpCAM x CD3 bi-specific antibody (Supplementary Fig S3). One alternative route of T cell activation is the STAT (signal transducer and activator of transcription) family of transcription factors which regulate cytokine expression in T cells (Yu et al, 2009). CCR9 expressed on MCF7 cells significantly

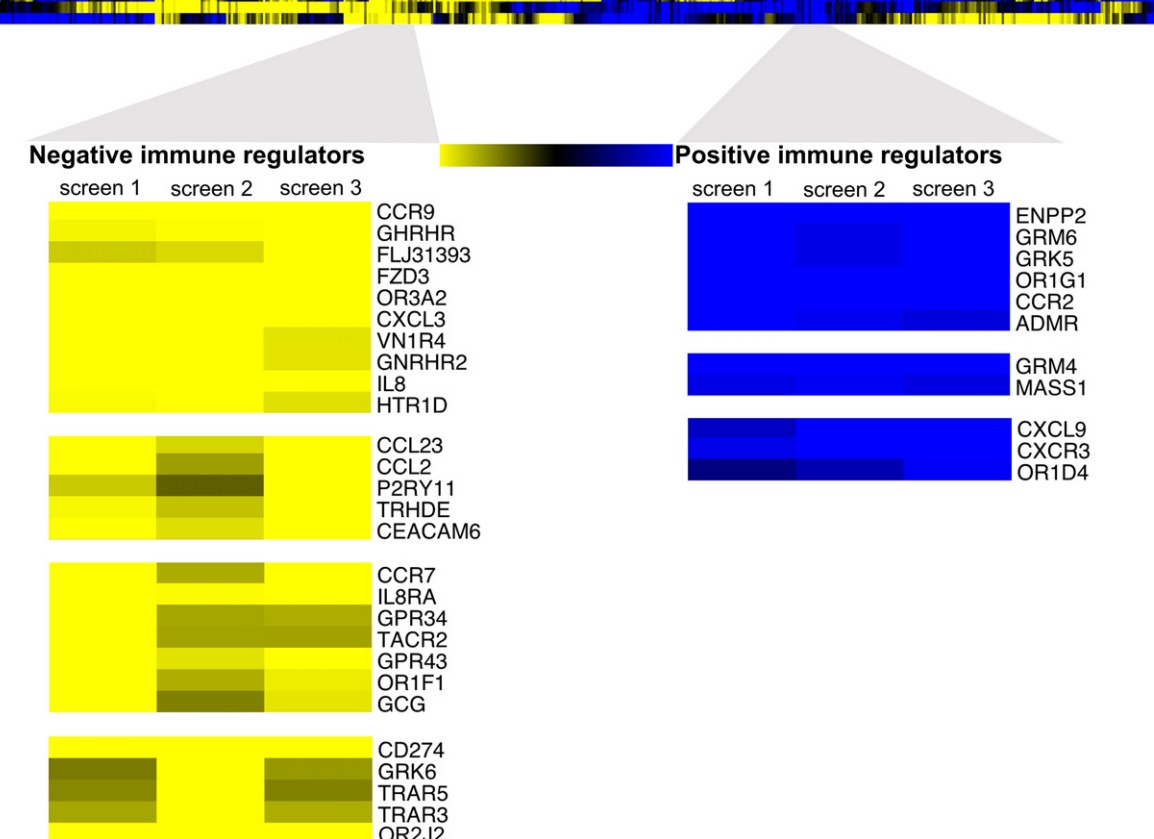

**Figure 3.   Heat map representation of potential positive and negative immune modulators identified from the RNAi screens.**
Differential scores were used to identify positive immune modulators (yellow) the knockdown of which enhance CTL-mediated cell killing and negative immune modulators (blue) the knockdown of which reduce CTL-mediated cell killing. Differential scores prior to filtering are shown for all genes tested in the 3 different screens (see Materials and Methods). Selected representative clusters of high-confidence hits are displayed herein.

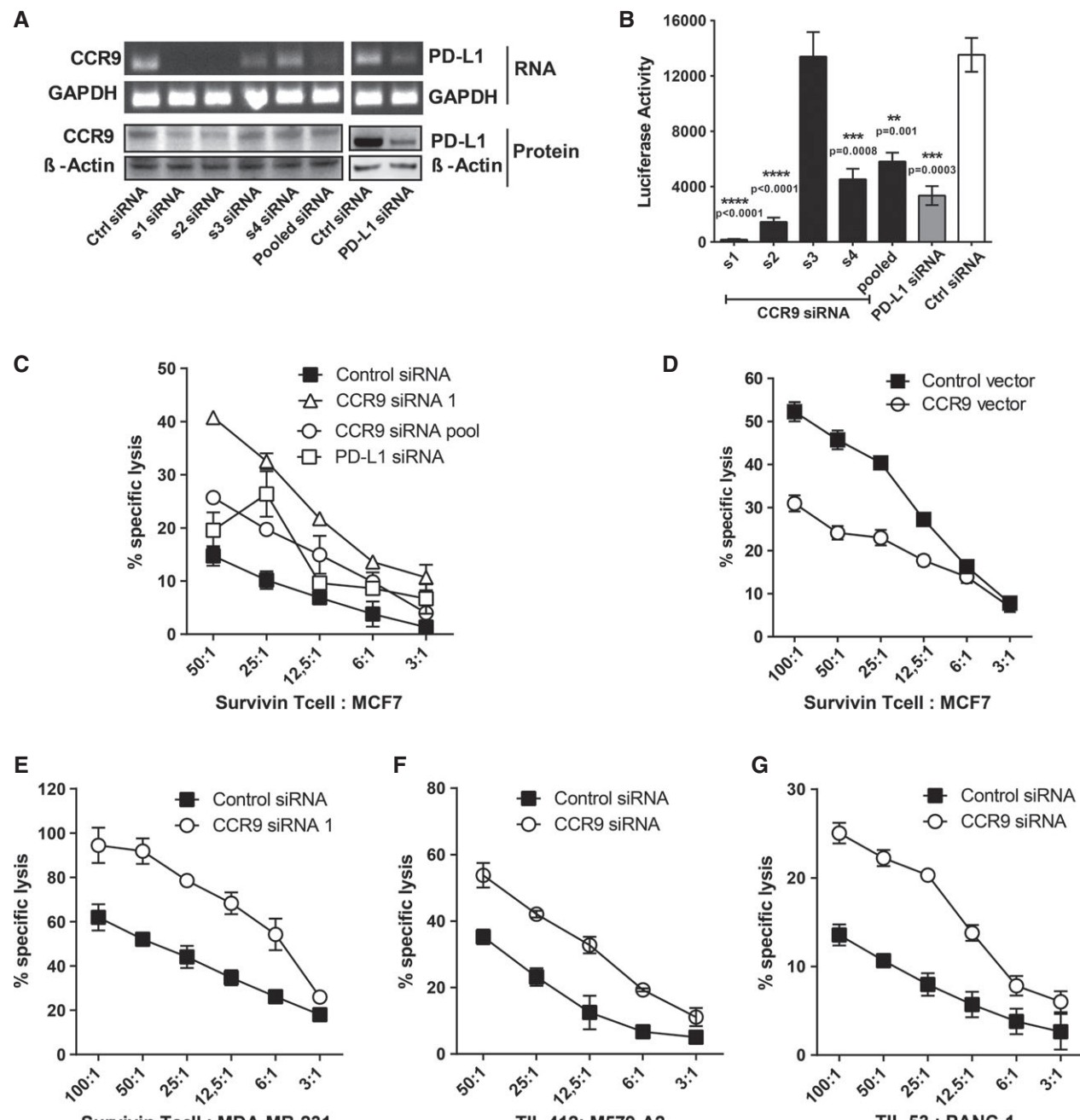

**Figure 4. CCR9 knockdown sensitizes tumor cells to immune attack.**

A    MCF7 cells were transfected with the described siRNA sequences and harvested after 72 h for mRNA and protein estimation using RT–PCR (upper) and immunoblot (lower) analysis, respectively. GAPDH and beta-actin were used as controls for RNA and protein normalization, respectively.

B    Luc-CTL cytotoxicity assay with PBMC-derived CTLs and bi-specific Ab as effector population and MCF7 as target cells, which were transfected with individual (s1–s4) or pooled CCR9 siRNA sequences. PD-L1 and non-specific control siRNAs were used as positive and negative controls, respectively, for CTL-mediated cytotoxicity.

C, D  Cr-release assay showing % specific lysis of MCF7 cells by survivin-specific T cells at different ratios upon CCR9 knockdown (C) or overexpression (D). MCF7 cells were transfected with either CCR9 siRNA s1 (Δ), pooled siRNA sequences (O), positive control PD-L1 (□), and non-specific control siRNA (■) (C) or with pCMV6-AC-His control vector (■) and pCMV6-AC-His-CCR9 expression construct (O) (D) 72 h prior to the assay.

E    Cr-release assay showing % specific lysis of MDA-MB-231 breast tumor cell line by survivin-specific T cells at different ratios upon CCR9 knockdown (O) in comparison to the control knockdown (■).

F, G  Cr-release assay showing lysis of patient-derived melanoma cells (M579-A2) by tumor-infiltrating lymphocytes (TIL 412) (F) or lysis of PANC-1 pancreatic adenocarcinoma cells by patient-derived pancreatic TIL 53 (G) at different E:T ratios upon CCR9 (O) or control (■) knockdown.

Data information: All experiments were performed in triplicates and are representative of at least three independent experiments. Error bars denote ± SEM, and statistical significance was calculated using the unpaired, two-tailed Student's *t*-test.

inhibited the secretion of the T-helper-1 (Th1) cytokines including tumor necrosis factor-alpha (TNF-α), interleukin-2 (IL-2), and (to a minor extent) of IFN-γ as well as IL-17, while the secretion of IL-10 was slightly but consistently increased (Fig 5C). Accordingly, we observed a significant increase in STAT1 and STAT2 signaling in survivin-specific T cells upon coculture with CCR9$^{lo}$ MCF7 cells, suggesting that anti-tumor type-1 immune response is impeded by tumor-specific CCR9 (Fig 5D and E).

Next, we assessed whether CCR9 expression in breast tumor cells affected T cell recognition directly or indirectly, for example, through CCR9 signaling-mediated increase in secretion of immune-suppressive factors. Since, the C-C chemokine ligand 25 (CCL25) is the only known interacting partner and ligand for CCR9, we first assessed whether CCL25 was involved in defining CCR9's tolerogenic phenotype. CCL25 was found to be produced by all the studied tumor cell lines, although at varied levels, as determined by ELISA (Fig 6A). Interestingly, shRNA-mediated stable knockdown of CCR9 did not affect CCL25 production by MCF7 breast cancer cells (Fig 6A). Next, inhibition of CCL25 using siRNAs (Fig 6B) or blocking antibody (Supplementary Fig S4A) showed no effect on antigen-specific lysis of MCF7 cells, in contrast to the CCR9 knockdown. However, it might still be possible that CCR9 mediates its immune-suppressive effect via other unknown soluble ligands or mediators. To examine this possibility, survivin-specific T cells were treated with the cell culture supernatants from either the CCR9 siRNA knockdown (CCR9$^{lo}$) or control (CCR9$^{hi}$) MCF7 tumor cells overnight and then challenged against CCR9$^{hi}$ or CCR9$^{lo}$ MCF7 cells in the cytotoxicity assay. Against the same tumor target, neither of the supernatant-treated T cells showed any difference in their recognition and lytic capacity. The difference in lysis between the different groups depended upon CCR9's expression on the tumor targets rather than on the T cell treatment (Fig 6C), hinting to the possibility that T cells can interact directly with CCR9 on tumor cells. To further assess whether intracellular signaling in tumor cells mediated by the surface-bound CCR9 plays any role in immunosuppression, pertussis toxin (PTX), a G$_{αi}$ inhibitor, was used. Although, pertussis toxin inhibited the migration of CCR9$^+$ tumor cells toward CCL25 in a transwell migration assay, proving its effectiveness in blocking CCR9's downstream signaling that is responsible for the chemotaxis (Supplementary Fig S4B), it, however, did not elicit elevated tumor lysis by antigen-specific T cells when compared to the CCR9 gene knockdown (Fig 6D). This further supported the notion that CCR9-mediated immune suppression on T cells might be independent of its intracellular signaling in the tumor cells and rather affects the T cells directly. Additionally, we evaluated whether CCR9 knockdown influences MHC-I expression on the tumor targets that could possibly explain their impact on T cell recognition and lysis. However, flow cytometric analysis revealed no major alterations in the surface expression of HLA-A2 on the target tumor cell lines upon CCR9 knockdown (Supplementary Fig S5).

To better understand the mode of CCR9-mediated immune suppression on T cells, we undertook broad-scale transcriptomics study to compare the changes in the transcriptome of T cells that encounter CCR9$^{hi}$ versus CCR9$^{lo}$ MCF7 tumor cells. Microarray analysis comparing these two T cell populations revealed a list of differentially up- and downregulated genes in CCR9$^{lo}$-treated T cells, which are represented in the volcano plot of Fig 6E and in the associated heat map of Fig 6F. Immune response-related genes such

as integrin alpha-2 (ITGA2; Yan et al, 2008), lymphotoxin alpha LTA; (Dobrzanski et al, 2004), interleukin 2 receptor alpha (IL2RA; Pipkin et al, 2010), and cytokine-inducible SH2-containing protein (CISH; Li et al, 2000) were upregulated, whereas genes that inhibit T cell maturation and effector function such as ephrin-A1 (EFNA1; Abouzahr et al, 2006), Kruppel-like factor 4 (KLF4; Wen et al, 2011), inhibitor of DNA binding-1 (ID1; Qi & Sun, 2004), transducer of ERBB2, 1 (TOB1; Tzachanis et al, 2001) were downregulated in T cells encountering CCR9$^{lo}$ tumor cells, which was found to be in accordance with the observed increase in cytotoxicity as shown before. Gene annotation/ontology (GO) analysis of the top upregulated genes revealed an enrichment of genes involved in positive regulation of immune response, while genes involved in lymphocyte maturation and apoptosis were enriched in the list of downregulated genes (Supplementary Fig S6A). We next wondered if these gene signatures observed in T cells upon tumor-specific CCR9 knockdown overlap with gene signatures generally associated with an activated T cell population. Using a publically available gene expression study comparing unstimulated CD8$^+$ T cells to CD3/CD28 antibody and IL-2-activated T cells (Wang et al, 2008), we indeed identified overlapping gene signatures in both these studies (Supplementary Fig S6B), suggesting that CCR9 knockdown on tumor cells favors better survival, proliferation, and activation of the encountering T cells.

Finally, to evaluate the in vivo relevance of CCR9 as a tumor-associated immunosuppressive entity, CCR9 was stably knocked down in the melanoma patient-derived M579-A2 tumor cell culture using CCR9-specific shRNA (shCCR9) or the control non-targeting shRNA (shControl; Supplementary Fig S7A). As expected, stable CCR9 knockdown tumor cell variants were more susceptible to immune lysis by melanoma patient-derived tumor-infiltrating lymphocytes (TIL 209) than their counterparts in the chromium-release cytotoxicity assay (Fig 7A), with no significant difference noted on the surface HLA-A2 expression upon CCR9 knockdown (Supplementary Fig S7B). For the in vivo analysis, $5 \times 10^5$ cells each of the CCR9$^+$ M579-A2 (shControl) and CCR9$^-$ M579-A2 (shCCR9) tumor cell lines were subcutaneously implanted in the left and the right flank, respectively, of the NSG immune-deficient mice (scheme in Fig 7B). These mice then received intravenous injection of $1 \times 10^7$ tumor-infiltrating lymphocytes (TIL 209) at Day 2 and Day 9. As shown in Fig 7C, CCR9$^-$ M579-A2 tumors grew significantly slower than the CCR9$^+$ tumors in response to the adoptive T cell transfer, indicating that CCR9 suppresses the anti-tumor activity of the transferred T cells in vivo as well. No difference in the tumor growth kinetic between the CCR9$^+$ and the CCR9$^-$ tumor cells was observed in mice that received no T cell transfer (Fig 7D). Taken together, these results suggest an important role for tumor-associated CCR9 as an immune-checkpoint node for application in cancer immunotherapy.

## Discussion

Here, we report a high-throughput screening strategy to comprehensively identify new cancer-associated immune-checkpoint molecules that promote immune resistance in tumors. Current state-of-the-art cancer immunotherapies—involving antigen-specific vaccines or adoptive cellular therapies with tumor-specific CTL—(Gao et al, 2013) are faced with the problem of cancer cell resistance to specific T cell attack (Rabinovich et al, 2007). We therefore chose

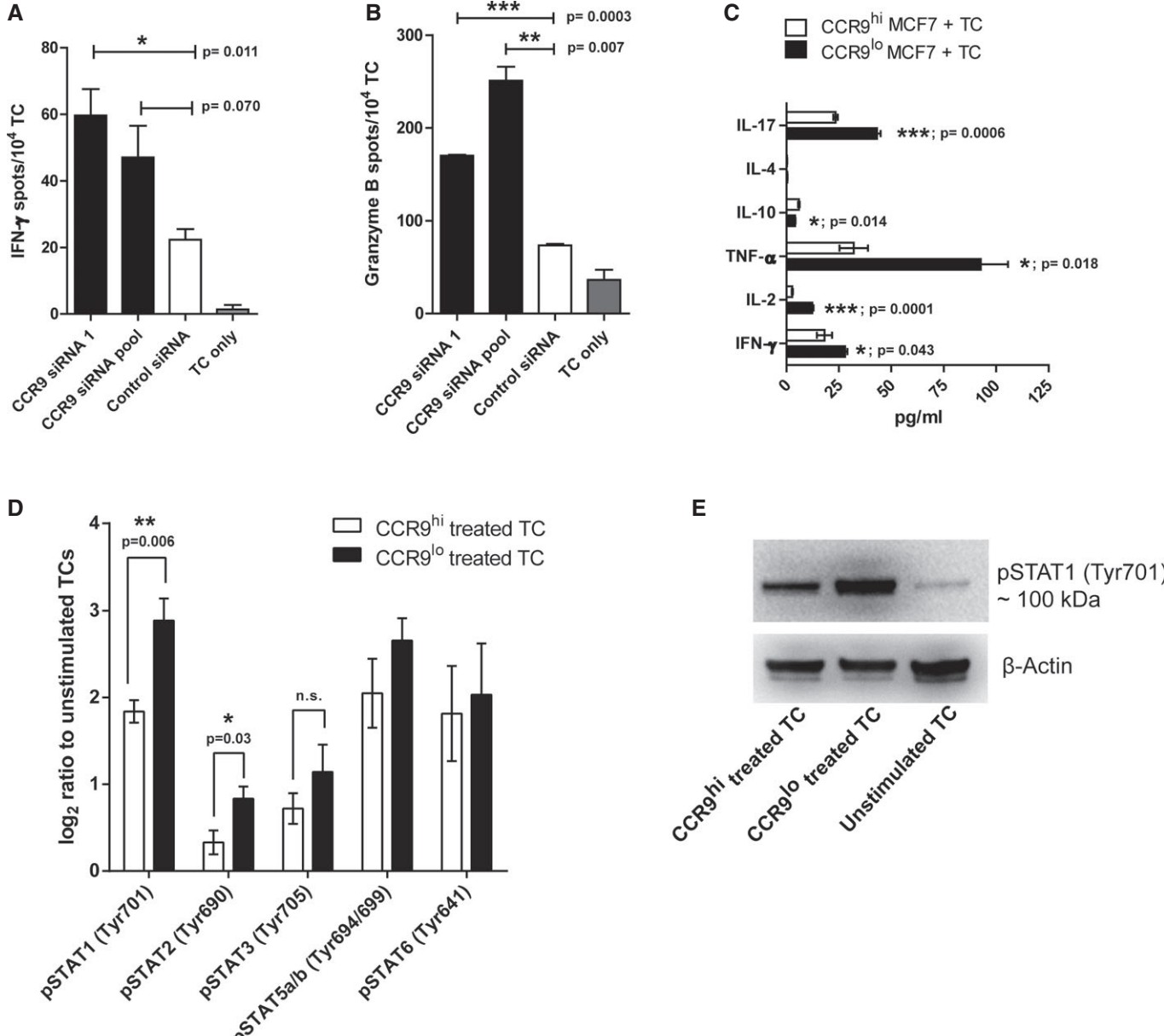

**Figure 5. Tumor-specific CCR9 impedes Th1-type immune response.**

A, B  ELISpot assay showing IFN-γ (A) and granzyme B (B) secretion by survivin-specific T cells, as spot numbers, upon CCR9 knockdown (black bars) in MCF7 cells compared to the control knockdown (white bars). T cells (TC) alone (grey bars) were used as control for background spot numbers.

C  Luminex assay showing cytokine levels in the supernatant from the coculture of survivin-specific TC and either CCR9hi MCF7 (transfected with CCR9-specific siRNA) or CCR9lo MCF7 (transfected with control siRNA) cells.

D  Phospho-plex analysis showing the phospho-STAT levels in survivin-specific TC upon encountering CCR9hi or CCR9lo MCF7 cells. Log2 ratio of mean fluorescent intensity (MFI) of the respective analytes to the unstimulated TC is plotted herein.

E  Immunoblot analysis showing the phospho-STAT1 levels in the CCR9hi-treated, CCR9lo-treated or unstimulated TC using the phospho-specific STAT1 (pTyr701) antibody. Beta-actin was used as the loading control.

Data information: In all the cases, experiments were performed in triplicate with at least two independent repeats. Mean ± SEM are shown herein, unless stated otherwise, with statistical significance assessed using unpaired, two-tailed Student's *t*-test.

Source data are available online for this figure.

CTL-induced tumor cell death as the key selection criterion for the screen. In this regard, our approach differs from a recent study that assessed the molecules involved in the inhibition of IFN-γ secretion by natural killer (NK) cells (Bellucci *et al*, 2012).

In our hands, the chosen luciferase-based determination of tumor cell death correlated well with classical cytotoxicity tests and was suitable for application in a high-throughput format. In order to avoid bias in hit identification based on single

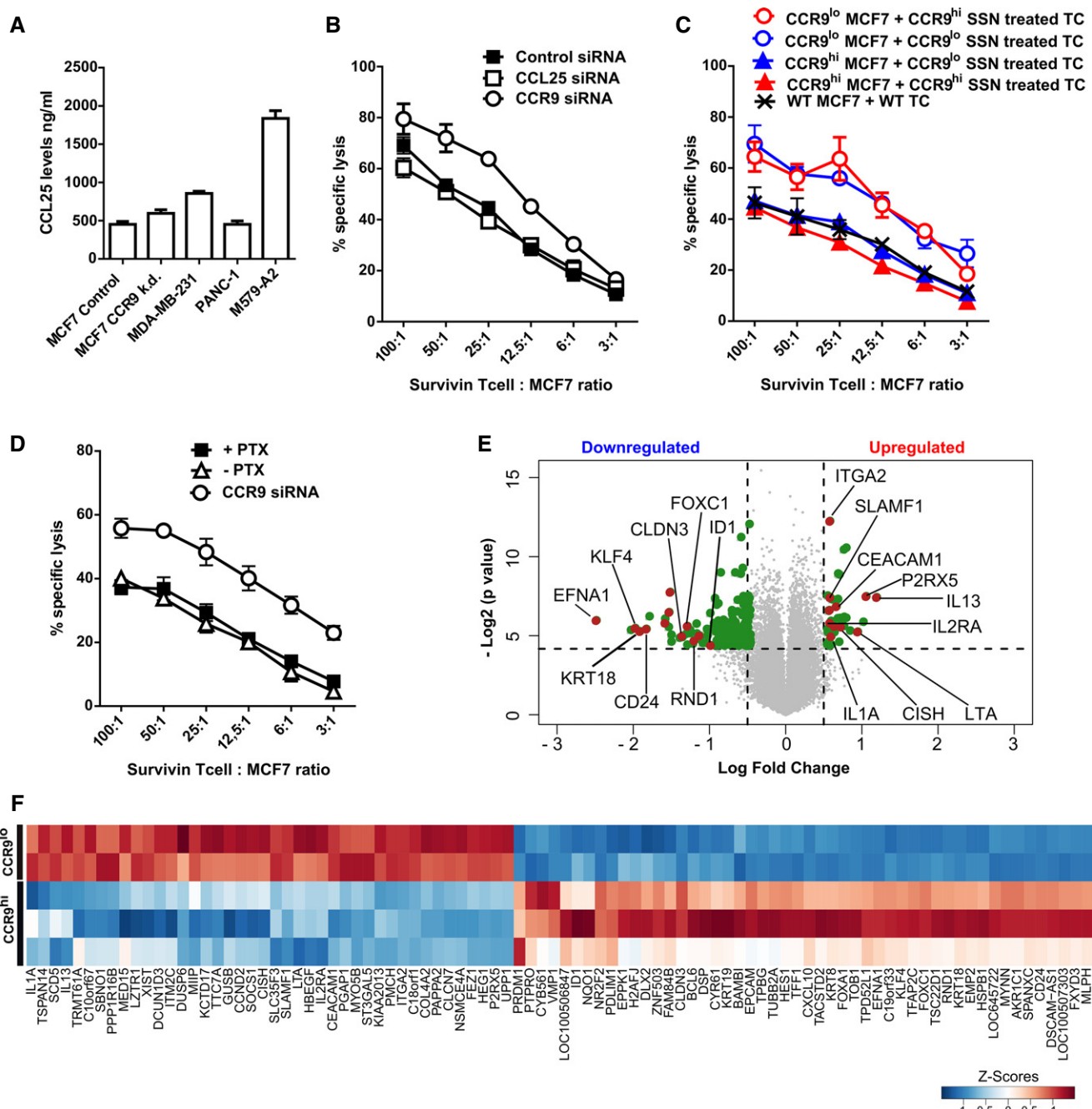

**Figure 6.   Tumor-specific CCR9 interacts directly with T cells inducing prominent changes in the gene expression signature.**

A   ELISA showing CCL25 levels in cell lysates from indicated tumor cell lines. CCR9 knockdown (k.d.) in MCF7 cells was achieved using specific shRNA (see Materials and Methods).

B   Cr-release assay showing % specific lysis of MCF7 cells by survivin TC upon CCL25 (□) or CCR9 (○) inhibition using specific siRNAs in comparison to the control siRNA (■). Mean ± SEM are depicted herein.

C   MCF7 cells were transfected with control or CCR9-specific siRNAs, and 48 h later, the supernatants (CCR9$^{lo}$ or CCR9$^{hi}$ SSN, respectively) were used to culture survivin TCs overnight. Supernatant-treated TCs were then used as effector cells against CCR9$^{lo}$ or CCR9$^{hi}$ MCF7 tumor cells in the Cr-release assay along with wild-type MCF7 cells. Mean ± SEM are depicted herein.

D   Cr-release assay showing % specific lysis of MCF7 cells that were pre-treated with or without pertussis toxin (PTX), or knocked down for CCR9 using specific siRNA. Mean ± SEM are depicted herein.

E, F   MCF7 cells transfected with control siRNA (CCR9$^{hi}$) or CCR9 siRNA (CCR9$^{lo}$) were cocultured with survivin TCs for 12 h. Gene microarray was performed with the total RNA extracted from purified T cells after the coculture. Volcano plot (E) illustrating fold change (FC; log2) in gene expression intensities compared with P-value (−log2) between CCR9$^{hi}$- and CCR9$^{lo}$-treated TCs. Horizontal bar at y = 4.32 represents a statistical significance of P = 0.05 (genes in gray below this line did not reach significance). LogFC cutoff at ± 0.5 is represented by the vertical lines. Heatmap representation of the top upregulated (LogFC > 0.5) and downregulated (LogFC < −0.85) genes (F) with P ≤ 0.05. Individual replicates per sample group are shown herein.

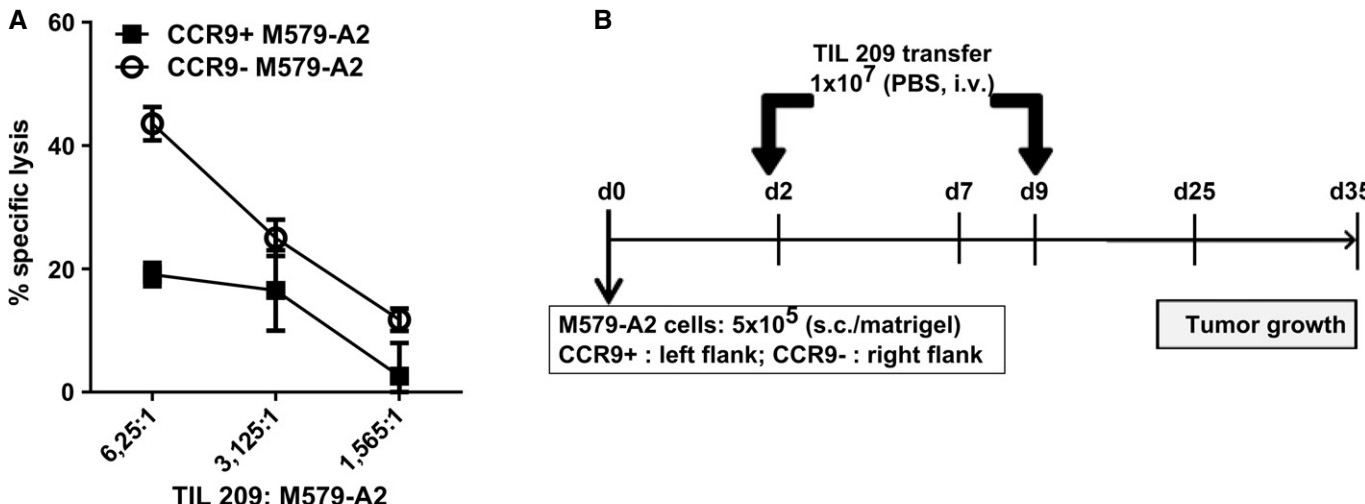

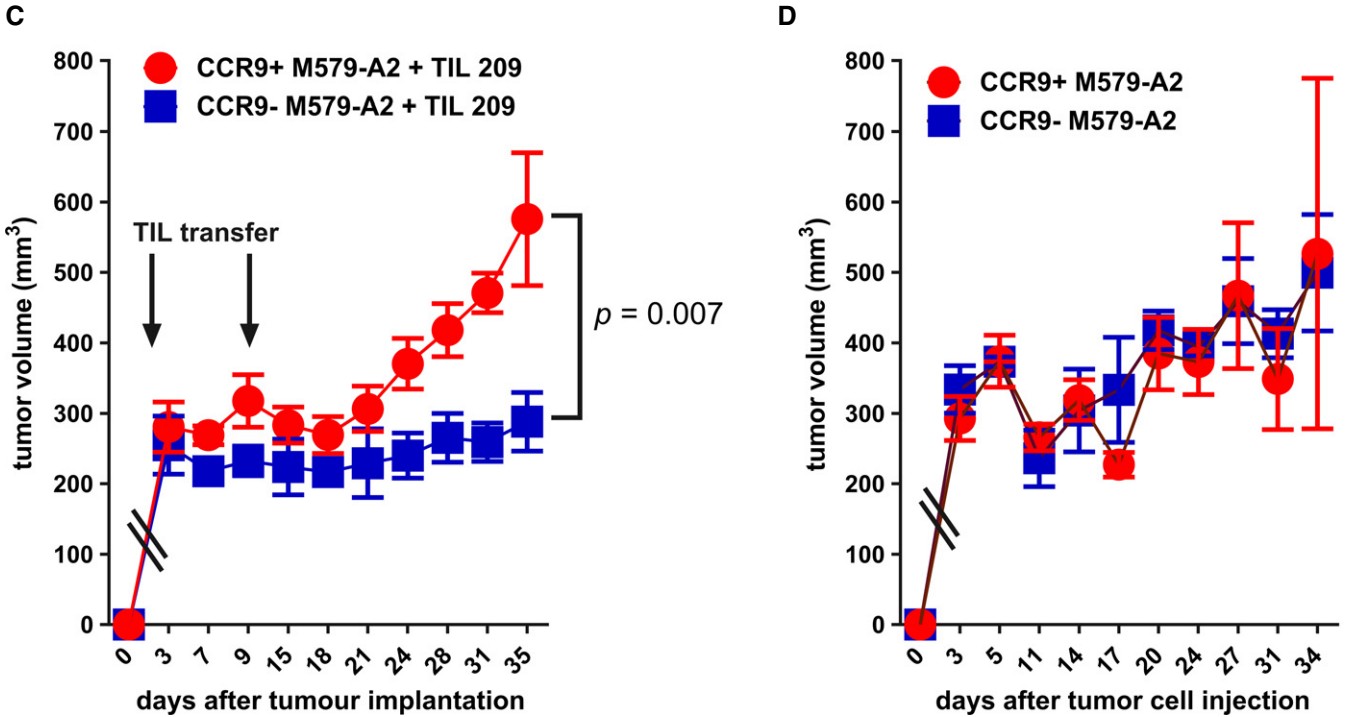

**Figure 7. *In vivo* inhibition of CCR9 significantly reduces tumor outgrowth in response to adoptive TIL therapy.**

A    Cr-release assay showing TIL 209-mediated lysis of CCR9⁺ M579-A2 (transduced with control shRNA) or CCR9⁻ M579-A2 cells (transduced with CCR9-specific shRNA). Curves represent mean ± SEM.

B    Scheme for the *in vivo* mouse experiment involving the s.c. injection of CCR9⁺ (shControl) or CCR9⁻ (shCCR9) M579-A2 tumor cells in the left and right flank, respectively, of the NSG mice. Following this, at d2 and d9, mice received i.v. injection of TIL 209 in PBS (*n* = 7) or PBS alone (control group for tumor growth; *n* = 3) and measured for tumor growth.

C, D  Tumor growth curves showing mean ± SEM tumor volume of CCR9⁺ or CCR9⁻ M579-A2 tumors in TIL-treated mice (C) or the PBS alone group (D). Statistical difference was calculated using the unpaired one-sided Mann–Whitney *U*-test.

donor-derived T cells, we performed the entire screen thrice, each time in duplicates, with both antigen-restricted as well as antigen-unrestricted approach. Thereby, we were able to obtain reproducible and overlapping candidates. By reliably confirming already established immune-modulatory ligands in breast cancer cells, the screening procedure proved its capacity to identify

immune-modulatory ligands on tumor cells. Based on the 520 genes tested in total, we observed a high hit rate for identifying potential immune modulators, especially when put in context of genome-wide screens and therefore warrants careful functional validation of individual candidates. However, immune response-related genes represent a major proportion of the genome and our focused

screening library was enriched in cell surface signaling proteins that might play a crucial role in immune modulation (Zhu et al, 2011). Moreover, a previous screen on NK cell-modulating ligands also revealed a high number of candidates (Bellucci et al, 2012). Thus, our high discovery rate might indeed represent complex regulatory networks governing peripheral immune surveillance. So far, we have performed experimental validation on four of the identified candidates (GLIPR1L1, GHSR, CEACAM6, and CCR9, of which we here report on CCR9) using additional tumor cell lines and antigen-specific CTL clones and in all cases we could confirm their immune-modulatory function. On the other hand, immune-activating ligands that facilitate anti-tumor activity of CTLs, such as CXCL9 which is known to correlate with higher immune infiltrate and better prognosis in colorectal cancer (Mlecnik et al, 2010), could also be uncovered from our screening methodology.

Furthermore, our screen revealed for the first time a role of CCR9 in T cell function. Tumor-specific knockdown of CCR9 resulted in increased activation of STAT signaling in T cells, in changes of gene expression that correlated with better and improved T cell effector functions, and in increased production of Th1 cytokines (IFN-γ, IL-2, and TNF-α) by antigen-specific CTLs, all of which have been shown to correlate with prolonged CTL survival and in vivo tumor rejection (Fallarino & Gajewski, 1999; Yu et al, 2009; Wilde et al, 2012). Although CCR9 did not induce strong changes in TCR signaling, a notable reduction in pan-Tyr phosphorylation was observed for Lck. Since Lck is activated via phosphorylation (Y394) and dephosphorylation (Y505) at both the tyrosine residues (Caron et al, 1992), it might be noteworthy to individually assess the impact of CCR9 upon these activating and repressive motifs of Lck. CCR9 knockdown also caused increased tumor lysis by CTL, which was reverted by CCR9 overexpression. Since the observed immune-modulatory effect was not based on CCR9-mediated tumor cell intrinsic signaling, nor mediated through soluble factors including CCL25, our data favor the assumption of a direct interaction between CCR9 on tumor cells and a yet unknown immune-modulatory ligand on T cells.

In our xenografted tumor mouse model, effective control of tumor outgrowth by the adoptively transferred TILs was only possible upon stable knockdown of CCR9 on the tumor cells in contrast to the CCR9+ tumor counterpart. These findings suggest that CCR9, which is expressed on many human tumors (Shen et al, 2009; Johnson-Holiday et al, 2011; Singh et al, 2011), plays an important role in tumor-immune evasion. Further development of CCR9 as an immunotherapeutic target for cancer treatment would, however, require extensive toxicological analysis. Previous study with CCR9 knockout mouse model has reported no adverse side effects of CCR9 gene ablation, besides the reduction in the γδ IELs which are characterized by their CCR9+ phenotype (Wurbel et al, 2001). However, given that CCR9 is important for the homing of effector T cells to the gut, systemic targeting of CCR9 would be a contraindication for gut-associated tumors. On the other hand, we found CCR9 to be expressed on only a small minority of survivin-specific T cells (~6%), while it was undetectable on peripheral blood T cells from healthy donors. Its expression in other immune cell subsets and that on healthy tissue remains to be examined.

Taken together, we here introduce an effective design of a high-throughput screen to uncover a broad panel of new immune-suppressor genes in breast cancer. In all, our methodology could enable the characterization of the 'immune modulatome' of tumor cells, which might facilitate the selection of target molecules for cancer immunotherapy.

## Materials and Methods

### Cell culture and reagents

MCF7, MDA-MB-231 (breast cancer), and PANC-1 pancreatic cancer cells were acquired from American Type Cell Culture (Wesel, Germany). MCF7luc cells were generated by electroporation with pEGFP-Luc plasmid and expansion of sorted GFP+ clones in selection medium containing 550 μg/ml G418 (Gibco, UK). M579-A2 melanoma culture was established from a patient and stably transfected with HLA-A2 expression construct as described before (Machlenkin et al, 2008). For stable CCR9 knockdown, lentiviral particles were produced using the pRSI9-U6-TagRFP-2APuro lentiviral expression vector (Cellecta) that contained either the CCR9-specific shRNA hairpin (ACCGGGCCAGTGGAGGTCTTTGTTCTGTTAATAT TCATA GCAGAACAAGGACCTTCACTGGCTTTT) or control non-targeting shRNA. Viruses were packaged using the psPAX2 and pMD2.G packaging plasmids (Addgene), and tumor cell lines were transduced with the viral particles as per the manufacturer's protocol.

For RNAi screens, CD8+ T cells were isolated from PBMC of healthy donors using CD8 Flow Comp kit (Invitrogen; Karlsruhe, Germany) and activated for 3 days in X-vivo medium (Lonza, Belgium) containing anti-CD3/CD28 activation beads (Dynal, Invitrogen) and 100 U/ml interleukin 2 (IL-2). HLA-A0201-restricted survivin$_{95-104}$ (clone SK-1) specific CTL clones were generated from PBMC of healthy donors as described (Brackertz et al, 2011). Tumor-infiltrating lymphocytes 412 and 209 microcultures were expanded from an inguinal lymph node of a melanoma patient as described (Dudley et al, 2010). TIL 53 microculture was established from a male patient with poorly differentiated pancreatic adenocarcinoma (PDAC) (Poschke & Offringa, unpublished data) and expanded using the rapid expansion protocol (REP) as described elsewhere (Dudley et al, 2003).

### RNAi screen and data analysis

The GPCR-targeting sub-library of the genome-wide siRNA library siGENOME (Dharmacon, GE Healthcare) contained 520 siRNA pools, consisting of four synthetic siRNA duplexes each and was prepared as described (Gilbert et al, 2011). Four RNAi screens were performed in duplicate wells. Positive and negative siRNA controls were distributed into empty wells prior to screening. Reverse siRNA transfection was performed by delivering 0.05 μl of RNAiMAX in 15 μl RPMI (Invitrogen). After 30 min, 3,000 MCF7 cells (screens 1 and 3: MCF7luc, screens 2 and 4: MCF7) in 30 μl DMEM medium (Invitrogen) supplemented with 10% FBS (Invitrogen) were added. Plates were incubated at 37°C for 24 h, and for screen 2, cells were transiently transfected with a luciferase expression plasmid (pEGFP-Luc) using TransIT-LT1 transfection reagent (Mirius Bio LLC, Madison, USA). 72 h post siRNA transfection, cancer cells were either challenged with CTLs and anti-EpCAM x CD3 bi-specific antibody (0.2 μg/well; screens 1 and 2) or survivin-specific CTLs (screen 3) or left untreated (condition without addition of CTLs and screen 4). Tumor lysis was quantified by analysis of residual luciferase expression in tumor cells (Brown et al, 2005). Screen 1 contained

CTLs from one single donor and screen 2 contained CTLs from 2 different donors; one for each technical replicate within the screen. 18 h later, supernatant was removed, cells were lysed, and luciferase measurements (screens 1, 2, and 3) or viability measurements using CellTiter-Glo (Promega) (screen 4) were performed as previously described (Muller *et al*, 2005; Gilbert *et al*, 2011). Plate reader data from RNAi screens were analyzed using the cellHTS2 package in R/Bioconductor (Boutros *et al*, 2006). Scores from both conditions, that is, addition of CTLs and without addition of CTLs, were quantile normalized against each other using the aroma.light package in R. Differential scores were calculated using a loess regression fitting. To reveal high-confidence hits, unsupervised hierarchical clustering of differential score of all genes from all screens was performed using the loess score. In order to robustly identify genes that positively modulate CTL-mediated cytotoxicity and to avoid biases potentially introduced by employing CTLs from different donors and employing genetically engineered as well as unmodified MCF7 cells, we filtered out genes that had a score > 2, and < −2 in the condition without addition of CTLs and had a score > 0.5, and < −0.5 in the condition with addition of CTLs. Finally, genes scoring in a CTG-based viability screen were filtered out from the candidate list (score < −1.5 and > 1.5). Thereby, siRNAs generally affecting cell viability, as determined by intracellular ATP levels, were excluded.

### Chromium-release cytotoxicity assay

Tumor cells were transfected with the described siRNAs using RNAi-MAX or with pCMV6-AC-His-CCR9 encoding vector and empty control vector (OriGene, Rockville, USA) using TransIT-LT1. 72 h later, transfected cells were harvested for chromium-release cytotoxicity assay as detailed in Supplementary Methods. For CCR9 blockade using pertussis toxin (PTX), $10^6$ tumor cells were incubated with 250 ng/ml of PTX (Sigma Aldrich) for 1 h at 37°C before labeling with radioactive chromium.

### ELISpot assay

IFN-γ and granzyme B secretion from T cells was determined using ELISpot assay as described by the manufacturer (Mabtech, Nacka Strand, Sweden) and detailed in the Supplementary Methods.

### Cytokine and phospho-plex analysis

Cytokines in T cell stimulation cultures were determined with Bio-Plex Pro Assay kit (Biorad, Germany). For phospho-TCR and phospho-STAT analysis, $2 \times 10^6$ survivin-specific TCs were cocultured with the respective target tumor cells at 20:1 ratio for defined time points, then isolated and lysed. Protein lysates were used for 7-plex TCR phosphoprotein kit and phospho-STAT 5-plex kit (Millipore, Billerica, USA) as detailed in the manufacturer's protocol. Measurements were performed using Luminex100 Bio-Plex System (Luminex, Austin, US; see also Supplementary Methods).

### Global gene expression analysis

For transcriptomic analysis, $2.5 \times 10^5$ MCF7 cells per group were reverse transfected with control or CCR9 s1 siRNA in 6-well plates

and cocultured with $5 \times 10^6$ survivin T cells after 72 h for an additional 12 h. Following co-incubation, TCs were purified using the anti-EpCAM antibody-coated mouse IgG beads (detailed in Supplementary Methods) and total RNA was isolated using the RNeasy Mini kit (Qiagen) as instructed by the manufacturer. Gene expression analysis was performed using the GeneChip Human Genome U133 Plus 2.0 Array (Affymetrix). Gene expression intensity was quantile normalized, and significant differences in the log fold change of gene expression between the CCR9hi- versus the CCR9lo-treated TCs were evaluated using the Welch's *t*-test. Top differentially up- and downregulated genes were plotted as heat maps using heatmap.2 function in R. Expression data can be accessed using the ArrayExpress database (www.ebi.ac.uk/arrayexpress) under accession number E-MTAB-3244. CCR9-induced gene expression signature was compared with a publically available gene expression dataset from a previous study (Wang *et al*, 2008), which compared CD8+ T cells from the peripheral blood of healthy donors before and after 24 h of activation with anti-CD3/CD28 antibody plus IL-2. The published dataset was retrieved from the Gene Expression Omnibus using the accession code GSE7572 and analyzed using standard methods in R.

### *In vivo* experiments

Appropriate approval for animal work was obtained from the regulatory authorities (Regierungspräsidium, Karlsruhe) before the start of the experiment. Four- to six-week-old female NSG mice were ordered from the Animal Core Facility at DKFZ, Heidelberg. Mice were subcutaneously injected with $5 \times 10^5$ cells (in 100 μl of matrigel per injection) of each CCR9− M579-A2 (transduced with CCR9-specific shRNA) and CCR9+ M579-A2 (transduced with non-targeting control shRNA) cell lines in the left and the right flank, respectively. Following this, at Day 2 and Day 9, 7 out of the 10 tumor-bearing mice received adoptive transfer of expanded TIL 209 cells intravenously into the tail vein ($1 \times 10^7$ cells/100 μl PBS/mouse). The remaining three mice were injected with PBS alone to assess tumor growth in the absence of adoptive TIL transfer. Tumor measurements were performed using a digital caliper (Carl Roth) at the indicated time points, and tumor volume was measured using the formula: volume = height*width*width*(π/3).

### Statistical evaluation

Differences between test and control groups were analyzed by two-sided Student's *t*-test. In all statistical tests, a *P*-value ≤ 0.05 was considered significant. Statistical difference between the tumor growth curves *in vivo* was assessed using the unpaired one-sided Mann–Whitney *U*-test.

See Supplementary Methods for further details.

**Supplementary information** for this article is available online: http://embomolmed.embopress.org

### Acknowledgements

We thank R. Hasse (LMU Munich) for the pEGFP-Luc plasmid, G. Moldenhauer (DKFZ) for the anti-EpCAM x CD3 bi-specific antibody, the DKFZ Genomics and Proteomics Core Facility for performing gene expression microarray, S. Jünger

**The paper explained**

**Problem**

The functional blockade of immune-suppressive molecules used by tumor cells to shut down tumor-immune control has lately revolutionized cancer therapy by resulting in immune-mediated rejection of otherwise treatment-refractory advanced cancers. Despite their crucial importance, only few immune-modulatory ligands on tumor cells have been identified. A comprehensive characterization of the 'immune modulatome' of cancer is therefore highly warranted.

**Results**

We successfully established a rapid high-throughput siRNA-based screening system that resulted in the identification of multiple surface-expressed ligands on cancer cells that inhibit T cell-mediated breast cancer cell rejection. One such ligand is CCR9, which inhibits cytotoxic T cell effector function through regulating STAT activation, which results in altered patterns of secreted effector cytokines, reduced cytotoxic activity, and impaired capacity to reject established human tumors *in vivo*.

**Impact**

We here introduce for the first time a method for the comprehensive identification of the 'immune modulatome' of cancer cells that prevents T cell-mediated tumor cell destruction. Applying this method to different tumor entities will rapidly increase the knowledge about therapeutically relevant immune-checkpoint molecules. One selected novel immune-modulatory ligand, CCR9, revealed strong potential for function-blocking therapies.

and C. Hartl (both DKFZ) for help with mouse work, and D. Egger-Adam (DKFZ) for helping with manuscript submission.

## Author contributions

PB, NK, MBo, and MBr conceived and designed the experiments. NK, MBr, TS, and LU performed the experiments. TM performed the *in vivo* experiments. CK, AS, HC, IP, RO, HB, RK, and AM contributed reagents/materials/ analysis tools. MBr and AKS analyzed the data. NK and PB wrote the manuscript.

## Conflict of interest

The authors declare that they have no conflict of interest.

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
