## [Review Process File · EMBO Molecular Medicine]

A high-throughput RNAi screen for detection of immune-checkpoint molecules that mediate tumor resistance to cytotoxic T lymphocytes

Nisit Khandelwal, Marco Breinig, Tobias Speck, Tillmann Michels, Christiane Kreutzer, Antonio Sorrentino, Ashwini Kumar Sharma, Ludmila Umansky, Heinke Conrad, Isabel Poschke, Rienk Offringa, Rainer König, Helga Bernhard, Arthur Machlenkin, Michael Boutros, Philipp Beckhove

Corresponding author: Philipp Beckhove, German Cancer Research Center DKFZ

Review timeline:

Submission date:	09 July 2014
Editorial Decision:	29 August 2014
Revision received:	12 December 2014
Editorial Decision:	08 January 2015
Revision received:	16 January 2015
Accepted:	19 January 2015

Transaction Report:

Editor: Roberto Buccione

1st Editorial Decision

29 August 2014

Thank you for the submission of your manuscript to EMBO Molecular Medicine. We are very sorry that it has taken so long to get back to you on your manuscript, also due to unfortunate overlap with the vacation period.

In this case we also experienced unusual difficulties in securing three willing and appropriate reviewers. As a further delay cannot be justified I have decided to proceed based on the two available consistent evaluations.

Both Reviewers find merits in your manuscript although they raise significant issues that require your action. I will not dwell into much detail as their comments are detailed. I would like, however, to highlight a few main points.

Reviewer 1, as you will see, lists a number of issues that essentially address claims/statements lacking sufficient experimental support. Most importantly, however, s/he laments the lack of any mechanistic insight into how CCR9 knockdown actually leads to improved tumour cell killing by T cells. I note that this Reviewer suggests a number of possibilities for you to act upon.

Reviewer 2, is quite positive, but does raise one crucial point. Specifically s/he notes that although your work is quite interesting, to make it impactful and translationally relevant some degree of in vivo validation would be required, for instance along the lines suggested by her/himself. I realize

that this is not a minor request, but I do agree that this would complete the circle on your work.

In conclusion, while publication of the paper cannot be considered at this stage, we would be pleased to consider a substantially revised submission, with the understanding that the Reviewers' concerns must be addressed with additional experimental data where appropriate and that acceptance of the manuscript will entail a second round of review.

Please note that it is EMBO Molecular Medicine policy to allow a single round of revision only and that, therefore, acceptance or rejection of the manuscript will depend on the completeness of your responses included in the next, final version of the manuscript.

As you know, EMBO Molecular Medicine has a "scooping protection" policy, whereby similar findings that are published by others during review or revision are not a criterion for rejection. However, I do ask you to get in touch with us after three months if you have not completed your revision, to update us on the status. Please also contact us as soon as possible if similar work is published elsewhere.

I look forward to seeing a revised form of your manuscript as soon as possible.

***** Reviewer's comments *****

Referee #1 (Comments on Novelty/Model System):

The observation that CCR9 is inhibitory is novel and the screening method is also relatively novel. The authors lack a mechanism of action and attempt to over-interpret some small changes in places to furnish one.

Referee #1 (Remarks):

This is a compelling study with an attractive methodology which demonstrates a role for CCR9 expressed by tumor cells in limiting the capacity of cytotoxic T cells to kill them. The major problem with this study is lack of specific mechanisms to explain this effect. Is the CCR9 effect primarily tumor intrinsic or extrinsic on T cells? The authors suggest a T cell effect with gene expression analysis but never prove these changes are causative or primary. How might a T cell interact with CCR9 if the mechanism is direct interaction with T cells? Does CCR9 downregulation increase sensitivity of tumor cells to apoptosis directly in a TCR-independent manner - what if you add anti-FAS antibody or purified Perforin and Granzyme - are they killed more easily absent CCR9? Does downregulation of CCR9 increase MHC I expression? Is CCR9 secreted or cleaved off the tumor and present in the supernatant where it is acting as a sink for CCL25? Do the tumors make CCL25 and signal through their own CCR9 in an autocrine fashion? The point is that some realistic explanation of how CCR9 knockdown leads to better killing by T cells is needed - at the very least definitive proof of whether CCR9 expression is protecting the tumor intrinsically or suppressing T cells extrinsically is needed. The cytokine and signaling changes seen in T cells suggest a role in direct T cell suppression but could also result from more productive interactions from a de-repressed tumor (i.e. they could be secondary).

The authors state on page 7 that in the Phospho-plex analysis they "observed no impact of CCR9 on TCR signaling"; however, there seem to be differences between high and low exposure to CCR9 in pLCK, and non-concentration (high vs low) impact on pCREB, pCD3e, and pERK. If the latter changes are a result solely of exposure to MCF7 cells independent of CCR9 that should be established with truly CCR9 deficient MCF7 cells (CRISPR/CAS knockout for example if a >95% siRNA knockdown is not possible) or by blockade of CCR9 with antibodies.

On page 8 the authors assert that "Indeed, CCR9 expressed on MCF7 cells selectively inhibited the secretion of T-helper-1 (Th1) cytokines such as IFN- γ , interleukin-2 (IL-2) and tumor necrosis factor-alpha (TNF- α), as well as IL-17, but increased the secretion of the immunosuppressive cytokine IL-10 (Fig. 5c)." The changes in IFN- γ and IL-10, although potentially

significant by T-test, do not appear biologically significant given that the "increase" in IL-10 appears to be about 2pg/mL which is biologically meaningless and outside the resolution range of the Biorad kit for IL-10. For IFN γ the "decrease" of perhaps 5-10pg/mL is at least within the actual resolution range of the Biorad kit, but is of questionable biologic significance.

Later in the same paragraph the authors continue "Accordingly, we observed a significant increase in STAT1 and slight increases in STAT2 and STAT5a/b signaling in survivin-specific T cells upon coculture with CCR9^{lo} MCF7 cells, indicating that Th1-type immune response is impeded by tumor-specific CCR9 (Fig. 5d,e)." While the STAT1 data is significant and supports their hypothesis, the STAT5 values are not significant and therefore cannot be meaningfully invoked as inhibition of a Th1 response. The STAT3 inhibition appears as though it may be significant on the other hand but is not discussed.

While exposure to CCR9^{hi} vs ^{lo} expressing MCF7 cells produces a differential gene expression profile in T cells which hints at more effective expansion via IL-2 activation and potentially better survival via CISH, the overall transcriptional profile does not support the authors statement that "these data indicate reprogramming of the T cells to an effector phenotype upon CCR9 knockdown"

Referee #2 (Comments on Novelty/Model System):

This paper describes a novel screening methodology to identify cell surface molecules on cancer cells involved in down-regulation of T cell responses. The screening method is novel, simple and reliable. Using this method, the authors have identified several novel molecules involved T cell functional down-regulation, notably of cytotoxic |T cell function. The molecule CCR9 has not previously been implicated as a cancer cell surface molecule involved in such suppression. Its role was investigated in detail, and found to be based on direct downregulation of T cell function. Moreover CCR9 was found to exert the same negative role on other cancer cell lines. This work can lead to novel approaches to intervention to promote cancer immunotherapy involving effector T cells.

Referee #2 (Remarks):

This paper describes a novel screening methodology to identify cell surface molecules on cancer cells involved in down-regulation of T cell responses. The screening method is novel, simple and reliable. Using this method, the authors have identified several novel molecules involved T cell functional down-regulation, notably of cytotoxic |T cell function. The molecule CCR9 has not previously been implicated as a cancer cell surface molecule involved in such suppression. Its role was investigated in detail, and found to be based on direct downregulation of T cell function. Moreover CCR9 was found to exert the same negative role on other cancer cell lines. This work can lead to novel approaches to intervention to promote cancer immunotherapy involving effector T cells.

The one obvious question is how important CCR9 is in the overall inhibition of T cell function in vivo. The authors can show this by blocking CCR9 with antibodies in vivo or by showing that CCR9 KO cells are better eradicated in vivo than the parental CCR9⁺ tumor cells. This would require demonstration that CCR9 fulfills the same role on murine cancer cells and T cells. Potential side effects of CCR9 blocking should also be investigated and discussed.

Reviewer #1

We thank the reviewer for the careful, critical and constructive evaluation of our study.

SPECIFIC COMMENTS

Query 1: Is the CCR9 effect primarily tumor intrinsic or extrinsic on T cells? The authors suggest a T cell effect with gene expression analysis but never prove these changes are causative or primary. How might a T cell interact with CCR9 if the mechanism is direct interaction with T cells? Does CCR9 down regulation increase sensitivity of tumor cells to apoptosis directly in a TCR-independent manner - what if you add anti-FAS antibody or purified Perforin and Granzyme - are they killed more easily absent CCR9? Does down regulation of CCR9 increase MHC I expression? Is CCR9 secreted or cleaved off the tumor and present in the supernatant where it is acting as a sink for CCL25? Do the tumors make CCL25 and signal through their own CCR9 in an autocrine fashion? The point is that some realistic explanation of how CCR9 knockdown leads to better killing by T cells is needed - at the very least definitive proof of whether CCR9 expression is protecting the tumor intrinsically or suppressing T cells extrinsically is needed. The cytokine and signaling changes seen in T cells suggest a role in direct T cell suppression but could also result from more productive interactions from a de-repressed tumor (i.e. they could be secondary).

Response: *The reviewer raises a very important question whether CCR9 represents a direct immune modulatory molecule by interacting with T cells or whether its role is indirect by modulating immunogenic features of tumor cells or their sensitivity to tumor cell attack. To clarify this question we conducted a series of new experiments that are now shown as new Figure 6 A-D and new Supplementary Figures S3, S4A-B and S5.*

First, we assessed whether CCR9 mediated immune suppression of T cells requires interactions through the TCR. To this end, we cocultured polyclonal CD8 T cells with CCR9 proficient or – deficient tumor cells in the presence or absence of an anti CD3-anti EpCAM bispecific antibody which crosslinks the TCR to the tumor cell and afterwards assessed IFN- γ secretion by the T cells. As shown in the new Supplementary Figure S3, CCR9 deficiency resulted in increased IFN- γ secretion only in the context of TCR stimulation. We reported this finding in the revised results section of the manuscript (page 7, second paragraph, lines 9-12).

Second, we assessed HLA expression in wt and CCR9 knock down variants of tumor cells used throughout the study and did not detect considerable differences. Representative data is now shown in the new Supplementary Figure S5 and reported in the revised version of the results section (page 9, lines 11-15). Second, we studied in MCF7 breast tumor cells a potential autocrine loop of CCR9-mediated CCL25 secretion by comparing CCL25 secretion in wt and CCR9 knock down cells and did not detect reduced levels of CCL25 after CCR9 knock down. These findings are reported in new Figure 6A and on page 8, second paragraph, lines 1-8. Furthermore, we assessed, whether CCL25 knock down would impact on T cell mediated tumor cell killing. While CCR9 knock down increased tumor cell lysis by T cells, CCL25 knock down or antibody blockade in tumor cells did not do so. The data demonstrate that CCR9 does not modulate T cell function indirectly through CCL25 and are shown in new Figure 6B, new Supplementary Figure S4A and reported on page 8, second paragraph, lines 8-10.

Third, we assessed, whether CCR9 mediates in tumor cells the production and secretion of immune suppressive factors which then may have indirectly affected the cytotoxic potential of T cells. To this end, we treated T cells with supernatants from CCR9 proficient or –deficient MCF7 cells and assessed their capacity to lyse CCR9+ or CCR9- tumor cells. As shown in new Figure 6C (and reported on page 8, lines 19-25 and page 9, lines 1-3) we found that CCR9 deficient tumor cells were efficiently lysed by T cells – irrespective of the supernatant used – while CCR9 proficient tumor cells were poorly lysed - again independent of the supernatant present. Thus, CCR9's immune modulating function is not mediated through secretion of CCL25 or other immune suppressive factors, suggesting a direct interaction with a yet unknown receptor on T cells.

In a fourth set of experiments we asked whether signaling through CCR9 in the tumor cells is required for the decreased cytotoxic activity of T cells which would favour the interpretation of an indirect effect on T cells (e.g. through upregulation of other immune modulatory surface molecules such as PDL1). We therefore treated CCR9 positive tumor cells with pertussis toxin which blocks CCR9 signaling in the tumor cells and compared the effect on T cell mediated lysis with that

obtained by CCR9 knock down. The results are shown in new Figure 6D, new Supplementary Figure S4B and demonstrate that CCR9 signaling in tumor cells has no impact on T cell mediated cytotoxicity. We reported these findings in the revised results section of the manuscript (page 9, lines 3-11).

Taken together, we can conclude from these additional experiments that CCR9 exerts a direct, tumor extrinsic effect on T cells, most likely through interaction with a yet undefined immune modulatory receptor on T cells. In addition to reporting these findings in the revised results section, we discussed them in the revised discussion section (page 13, second paragraph, lines 7-10).

Query 2: The authors state on page 7 that in the Phospho-plex analysis they "observed no impact of CCR9 on TCR signaling"; however, there seem to be differences between high and low exposure to CCR9 in pLCK, and non-concentration (high vs low) impact on pCREB, pCD3e, and pERK. If the latter changes are a result solely of exposure to MCF7 cells independent of CCR9 that should be established with truly CCR9 deficient MCF7 cells (CRISPR/CAS knockout for example if a >95% siRNA knockdown is not possible) or by blockade of CCR9 with antibodies.

Response: The results regarding CCR9-dependent TCR signaling are shown in the Supplementary Figure S2. Here, we compared the response of a survivin-specific T cell clone towards CCR9 proficient or deficient MCF7 cells. As a positive control we stimulated the T cells polyclonally with PMA/Ionomycin instead of exposure to tumor cells. As polyclonal stimulation with PMA/Ionomycin should activate the T cells much stronger than MCF7 cells, we expected to see differences between the positive control and the test samples. This was indeed the case for pCREB, pCD3 and pERK. However, these factors were not found to be differentially activated after exposure to CCR9 proficient or deficient tumor cells. Regarding pLck we found a reduced activation after 5 minutes of exposure to CCR9 low tumor cells, while no differences could be seen after 1 or 20 minutes.

Taken together, these data do not strongly support the conclusion that TCR signaling is a major target of CCR9 mediated immune suppression. However, we agree that the difference in Lck activation is notable and therefore we revised the description of the respective findings in the results section as follows (page 7, second paragraph, lines 6-9): "With the exception of some degree of reduced Lck activation (which was detectable only 5 minutes after exposure to CCR9lo tumor cells), we did not observe any CCR9-dependent changes in TCR signaling". As the reviewer's request to some extent may have been caused by a misunderstanding of the figure labels, we relabeled the legend in the Supplementary Figure S2.

Query 3: On page 8 the authors assert that "Indeed, CCR9 expressed on MCF7 cells selectively inhibited the secretion of T-helper-1 (Th1) cytokines such as IFN- γ , interleukin-2 (IL-2) and tumor necrosis factor-alpha (TNF- α), as well as IL-17, but increased the secretion of the immunosuppressive cytokine IL-10 (Fig. 5c)." The changes in IFN-g and IL-10, although potentially significant by T-test, do not appear biologically significant given that the "increase" in IL-10 appears to be about 2pg/mL which is biologically meaningless and outside the resolution range of the Biorad kit for IL-10. For IFNg the "decrease" of perhaps 5-10pg/mL is at least within the actual resolution range of the Biorad kit, but is of questionable biologic significance.

Response: We agree that the change in IL-10 secretion is minor. However, the IL-10 reduction was reproducibly significant. We used only low numbers (10,000/well) of T cells in the assay which may explain the low absolute cytokine concentrations and the supposedly low absolute differences between the test groups. In a relative scale, IL-10 was consistently reduced by 30% and IFN- γ increased by 35%. We are therefore hesitant to simply ignore these changes, but agree to emphasize on the only weak differences for these two cytokines in the revised results section as follows (page 8 lines 3-6): "CCR9 expressed on MCF7 cells significantly inhibited the secretion of the T-helper-1 (Th1) cytokines including tumor necrosis factor-alpha (TNF- α), interleukin-2 (IL-2) and (to a minor extent) of IFN- γ as well as IL-17, while the secretion of IL-10 was slightly but consistently increased (Figure 5C)".

Query 4: Later in the same paragraph the authors continue "Accordingly, we observed a significant increase in STAT1 and slight increases in STAT2 and STAT5a/b signaling in survivin-specific T cells upon coculture with CCR9lo MCF7 cells, indicating that Th1-type immune response is impeded by tumor-specific CCR9 (Fig. 5d,e)." While the STAT1 data is significant and supports

their hypothesis, the STAT5 values are not significant and therefore cannot be meaningfully invoked as inhibition of a Th1 response. The STAT3 inhibition appears as though it may be significant on the other hand but is not discussed.

Response: Regarding STAT5 we agree with the reviewer and changed the text accordingly as follows (page 8 lines 6-9): "Accordingly, we observed a significant increase in STAT1 and STAT2 signaling in survivin-specific T cells upon coculture with CCR9^{lo} MCF7 cells, suggesting that anti-tumor type-1 immune response is impeded by tumor-specific CCR9 (Figure 5D, E)". Regarding STAT3, there was no significant difference ($p=0.28$) between the group and this has been now indicated by "n.s." on the associated graph in Figure 5D.

Query 5: While exposure to CCR9^{hi} vs ^{lo} expressing MCF7 cells produces a differential gene expression profile in T cells which hints at more effective expansion via IL-2 activation and potentially better survival via CISH, the overall transcriptional profile does not support the authors statement that "these data indicate reprogramming of the T cells to an effector phenotype upon CCR9 knockdown"

Response: We agree with the reviewer that our statement is not sufficiently supported by the gene expression data. We now rephrased the paragraph as follows (page 9, second paragraph, lines 6-10 and page 10 lines 1-12): "Immune response-related genes such as integrin alpha-2 (ITGA2) (24), lymphotoxin alpha (LTA) (25), interleukin 2 receptor alpha (IL2RA) (26), cytokine inducible SH2-containing protein (CISH) (27) were upregulated; whereas genes that inhibit T cell maturation and effector function such as ephrin-A1 (EFNA1) (28), Kruppel-like factor 4 (KLF4) (29), inhibitor of DNA binding-1 (ID1) (30), transducer of ERBB2, 1 (TOB1) (31) were downregulated in T cells encountering CCR9^{lo} tumor cells, which was found to be in accordance with the observed increase in cytotoxicity as shown before. Gene-annotation/ontology (GO) analysis of the top up-regulated genes revealed an enrichment of genes involved in positive regulation of immune response, while genes involved in lymphocyte maturation and apoptosis were enriched in the list of downregulated genes (Supplementary Figure S6A). We next wondered if these gene signatures associated with re-activated T cells upon tumor-specific CCR9 knockdown overlap with gene signatures generally associated with an activated T cell population. Using a publically available gene expression study comparing unstimulated CD8⁺ T cells to CD3/CD28 antibody and IL-2 activated T cells (32), we indeed identified overlapping gene signatures in both these studies (Supplementary Figure S6B), suggesting that CCR9 knockdown on tumor cells favors better survival, proliferation and activation of the encountering T cells".

Reviewer #2:

We thank the reviewer for careful consideration of our manuscript and invaluable suggestions for improvement.

Remarks: This paper describes a novel screening methodology to identify cell surface molecules on cancer cells involved in down-regulation of T cell responses. The screening method is novel, simple and reliable. Using this method, the authors have identified several novel molecules involved T cell functional down-regulation, notably of cytotoxic T cell function. The molecule CCR9 has not previously been implicated as a cancer cell surface molecule involved in such suppression. Its role was investigated in detail, and found to be based on direct down regulation of T cell function. Moreover CCR9 was found to exert the same negative role on other cancer cell lines. This work can lead to novel approaches to intervention to promote cancer immunotherapy involving effector T cells.

Query 1: The one obvious question is how important CCR9 is in the overall inhibition of T cell function in vivo. The authors can show this by blocking CCR9 with antibodies in vivo or by showing that CCR9 KO cells are better eradicated in vivo than the parental CCR9⁺ tumor cells. This would require demonstration that CCR9 fulfills the same role on murine cancer cells and T cells.

Response: We thank the reviewer for this important request. We have meanwhile conducted tumor rejection experiments in vivo using CCR9 proficient and -deficient variants of the melanoma cell line M579 which we generated through stable shRNA expression using lentiviral transduction. We now show in the new Figure 7A-D that CCR9 wt tumors cannot be controlled by adoptively

transferred tumor-specific TIL, while tumor growth of the CCR9 deficient counterparts is efficiently inhibited after TIL transfer.

Thus, CCR9 expression by tumor cells efficiently inhibits T cell based anti-tumor immunotherapy in vivo. These findings are reported in the revised results and discussion section (page 10, second paragraph; page 11, lines 1-5 and page 13, third paragraph, line 1-3).

Query 2: Potential side effects of CCR9 blocking should also be investigated and discussed.

Response: *Based on available literature on a CCR9 knockout mouse model we have now added a comment on potential side effects of CCR9 knock down in the revised discussion section as follows (page 13, second paragraph, lines 5-9 and page 14, lines 1-4): "Further development of CCR9 as an immunotherapeutic target for cancer treatment would however require extensive toxicological analysis. Previous study with CCR9 knockout mouse model has reported no adverse side effects of CCR9 gene ablation, besides the reduction in the $\gamma\delta$ IELs which are characterized by their CCR9+ phenotype (42). However, given that CCR9 is important for the homing of effector T cells to the gut, systemic targeting of CCR9 would be a contraindication for gut-associated tumors. On the other hand, CCR9 was found to be expressed on only a small minority of survivin-specific T cells (~ 6%) and was undetectable on peripheral blood T cells from healthy donors (data not shown). Its expression in other immune cell subsets and that on healthy tissue remains to be examined".*

2nd Editorial Decision

08 January 2015

Thank you for the submission of your revised manuscript to EMBO Molecular Medicine. We have now received the enclosed reports from the referees that were asked to re-assess it.

As you will see the reviewers are now globally supportive and I am pleased to inform you that we will be able to accept your manuscript pending the following final amendments:

- 1) Reviewer 1 provides some final comments that, although not requiring additional work at this time, in my opinion deserve to be at least discussed in the manuscript. The point s/he makes about the two tyrosine phosphorylation sites in Lck with opposing functions is especially relevant. Ideally, should you have any further data that would clarify the issue, I would suggest incorporating them. Should you not, perhaps a comment should be provided.
- 2) As per our Author Guidelines, the description of all reported data that includes statistical testing must state the name of the statistical test used to generate error bars and P values, the number (n) of independent experiments underlying each data point (not replicate measures of one sample), and the actual P value for each test (not merely 'significant' or 'P < 0.05').
- 3) I note that your manuscript contains a section called "Paper Summarized". This needs to be changed. Specifically we would need two separate items from you:
 - a) The Paper Explained - EMBO Molecular Medicine articles are accompanied, within the manuscript itself, by a structured summary of the article to emphasize the major findings of the paper and their medical implications for the non-specialist reader. Please provide a summary accessible to non-specialists and specialists alike, highlighting the medical issue you are addressing (heading: PROBLEM), the results obtained (heading: RESULTS), and their clinical impact (heading: IMPACT). This may be edited to ensure that readers understand the significance and context of the research. Please refer to any of our published primary research articles for an example.
 - b) Synopsis - Every published paper now includes a 'Synopsis' to further enhance discoverability. Synopses are displayed on the journal webpage and are freely accessible to all readers. They include a short standfirst (to be written by the editor) as well as 2-5 one sentence bullet points that summarise the paper (to be written by the author). Please provide the short list of bullet points that summarise the key NEW findings. The bullet points should be designed to be complementary to the abstract - i.e. not repeat the same text. We encourage inclusion of key acronyms and quantitative information. Please use the passive voice. Please attach these in a separate file or send them by

email, we will incorporate them accordingly.

3) We are now encouraging the publication of source data, particularly for electrophoretic gels and blots, with the aim of making primary data more accessible and transparent to the reader. Would you be willing to provide a PDF file per figure that contains the original, uncropped and unprocessed scans of all or at least the key gels used in the manuscript? The PDF files should be labeled with the appropriate figure/panel number, and should have molecular weight markers; further annotation may be useful but is not essential. The PDF files will be published online with the article as supplementary "Source Data" files. If you have any questions regarding this just contact me.

4) Data of gene expression experiments described in submitted manuscripts should be deposited in a MIAME-compliant format with one of the public databases. We would therefore ask you to submit your microarray data to the ArrayExpress database maintained by the European Bioinformatics Institute for example. ArrayExpress allows authors to submit their data to a confidential section of the database, where they can be put on hold until the time of publication of the corresponding manuscript. Please see <http://www.ebi.ac.uk/arrayexpress/Submissions/> or contact the support team at arrayexpress@ebi.ac.uk for further information.

5) Could you please re-phrase the "Mouse Work" subheading in the methods section? For instance, "Animal experimentation" or similar would be better.

6) Please provide a conflict of interest declaration

Please submit your revised manuscript within two weeks. I look forward to seeing a revised form of your manuscript as soon as possible.

***** Reviewer's comments *****

Referee #1 (Remarks):

I thank the authors for their diligent efforts in addressing the prior critiques.

While no further work is required. It is worth considering that functional increases in IL-10 should be accompanied by increased pSTAT3 which is not observed.

Also, the Lck result is interesting and may point to a more detailed mechanism. Your assay looked at pan-Tyr phosphorylation for Lck but Lck has both a repressive (Y505) and activating tyrosine (Y394) - it would be helpful to know which of these is being modulated to assess activation versus repression. I only know of one molecule that represses both Lck and IFN γ - Cbl-b - might be worth a look.

Referee #2 (Remarks):

The authors have adequately responded to the points raised and performed additional experiments with clearcut outcomes. This work can lead to novel biomarker and therapy insights associated with immunotherapy of cancer

2nd Revision - authors' response

16 January 2015

Thank you very much for re-evaluating our revised manuscript # EMM-2014-04414-V2 and for accepting it for publication in your journal. We have made the requested amendments accordingly,

which are underlined throughout the main manuscript file for better visibility. The changes made refer to the following:

1. While Reviewer #1's comments on regulation of IL-10 production by STAT3 signaling is a valid point, it does not however paint a complete picture as IL-10 production by T lymphocytes is known to be regulated by several other factors including SP1, SP3, C/EBP β , IRF1, MAF, Notch (referenced below). An insight into which signaling cascade specifically regulates IL-10 production in our T cells that encounter CCR9^{lo} tumor cells is beyond the scope of this study and we believe that an explanation into that direction in the Discussion section would divert the attention of the readers away from the main focus of this presented study. We therefore would recommend leaving our observations on changes in the IL-10 production as it is. On the other hand, the remark of the Reviewer regarding two opposing Tyr phosphorylation sites in Lck is well noted and we have accordingly adapted our discussion section to include the following statement (*page 13, second paragraph, lines 6-10*): *“Although CCR9 did not induce strong changes in TCR signaling, a notable reduction in pan-Tyr phosphorylation was observed for Lck. Since Lck is activated via phosphorylation (Y394) and dephosphorylation (Y505) at both the tyrosine residues (38), it might be noteworthy to individually assess the impact of CCR9 upon these activating and repressive motifs of Lck”*.

Tone, M., Powell, M. J., Tone, Y., Thompson, S. A. & Waldmann, H. IL-10 gene expression is controlled by the transcription factors Sp1 and Sp3. *J. Immunol.* 165, 286–291 (2000).

Brenner, S. et al. cAMP-induced interleukin-10 promoter activation depends on CCAAT/enhancer-binding protein expression and monocytic differentiation. *J. Biol. Chem.* 278, 5597–5604 (2003).

Ziegler-Heitbrock, L. et al. IFN- α induces the human IL-10 gene by recruiting both IFN regulatory factor 1 and Stat3. *J. Immunol.* 171, 285–290 (2003).

Saraiva, M. et al. Interleukin-10 production by Th1 cells requires interleukin-12-induced STAT4 transcription factor and ERK MAP kinase activation by high antigen dose. *Immunity* 31, 209–219 (2009).

Rutz, S. et al. Notch regulates IL-10 production by T helper 1 cells. *Proc. Natl Acad. Sci. USA* 105, 3497–3502 (2008).

2. In accordance with the author guidelines, we have now included the exact p-values for all the statistical tests in the figure panel itself, along with the description of statistical tests performed and independent experimental repeats in the corresponding figure legends. As an exception, in Figure 4B, where p values were found to be less than 0.0001 till the 7th decimal point, they have been highlighted with ‘**** p<0.0001’.
3. The ‘paper summarized’ section has now been replaced with the ‘Paper Explained’ part according to the guidelines. This could be found in the main manuscript file just before the Reference section. Synopsis for the paper is attached separately with this submission
4. We could provide the source data file on the key blots in the manuscript which is now attached as a separate PDF file. This concerns Figure 5E.
5. Gene expression dataset has now been submitted to the ArrayExpress database as recommended, with the accession number now included in the methodology section (page 18, line 8-10). Upon confirmation of online publication date of the article, we would ask

the curators to make the link public. Should you require 'Reviewer Access' to the dataset before online publication, please let us know so that we could provide you with a special login and password.

6. 'Mouse work' section has now been rephrased as '*In vivo* experiments' in the methods section.
7. The manuscript file now contains the conflict of interest statement (page 20).

We hope that this addresses the needs of the journal for publication, but should the editorial team require any further clarification then please do not hesitate to get in touch with us. We look forward to receiving the proof-read version of the manuscript soon.

Reviewer #1

We thank the reviewer for the careful, critical and constructive evaluation of our study.

SPECIFIC COMMENTS

Query 1: Is the CCR9 effect primarily tumor intrinsic or extrinsic on T cells? The authors suggest a T cell effect with gene expression analysis but never prove these changes are causative or primary. How might a T cell interact with CCR9 if the mechanism is direct interaction with T cells? Does CCR9 down regulation increase sensitivity of tumor cells to apoptosis directly in a TCR-independent manner - what if you add anti-FAS antibody or purified Perforin and Granzyme - are they killed more easily absent CCR9? Does down regulation of CCR9 increase MHC I expression? Is CCR9 secreted or cleaved off the tumor and present in the supernatant where it is acting as a sink for CCL25? Do the tumors make CCL25 and signal through their own CCR9 in an autocrine fashion? The point is that some realistic explanation of how CCR9 knockdown leads to better killing by T cells is needed - at the very least definitive proof of whether CCR9 expression is protecting the tumor intrinsically or suppressing T cells extrinsically is needed. The cytokine and signaling changes seen in T cells suggest a role in direct T cell suppression but could also result from more productive interactions from a de-repressed tumor (i.e. they could be secondary).

Response: *The reviewer raises a very important question whether CCR9 represents a direct immune modulatory molecule by interacting with T cells or whether its role is indirect by modulating immunogenic features of tumor cells or their sensitivity to tumor cell attack. To clarify this question we conducted a series of new experiments that are now shown as new Figure 6 A-D and new Supplementary Figures S3, S4A-B and S5.*

First, we assessed whether CCR9 mediated immune suppression of T cells requires interactions through the TCR. To this end, we cocultured polyclonal CD8 T cells with CCR9 proficient or – deficient tumor cells in the presence or absence of an anti CD3-anti EpCAM bispecific antibody which crosslinks the TCR to the tumor cell and afterwards assessed IFN- γ secretion by the T cells. As shown in the new Supplementary Figure S3, CCR9 deficiency resulted in increased IFN- γ secretion only in the context of TCR stimulation. We reported this finding in the revised results section of the manuscript (page 7, second paragraph, lines 9-12).

Second, we assessed HLA expression in wt and CCR9 knock down variants of tumor cells used throughout the study and did not detect considerable differences. Representative data is now shown in the new Supplementary Figure S5 and reported in the revised version of the results section (page 9, lines 11-15). Second, we studied in MCF7 breast tumor cells a potential autocrine loop of CCR9-mediated CCL25 secretion by comparing CCL25 secretion in wt and CCR9 knock down cells and did not detect reduced levels of CCL25 after CCR9 knock down. These findings are reported in new Figure 6A and on page 8, second paragraph, lines 1-8. Furthermore, we assessed, whether CCL25 knock down would impact on T cell mediated tumor cell killing. While CCR9 knock down increased

tumor cell lysis by T cells, CCL25 knock down or antibody blockade in tumor cells did not do so. The data demonstrate that CCR9 does not modulate T cell function indirectly through CCL25 and are shown in new Figure 6B, new Supplementary Figure S4A and reported on page 8, second paragraph, lines 8-10.

Third, we assessed, whether CCR9 mediates in tumor cells the production and secretion of immune suppressive factors which then may have indirectly affected the cytotoxic potential of T cells. To this end, we treated T cells with supernatants from CCR9 proficient or -deficient MCF7 cells and assessed their capacity to lyse CCR9⁺ or CCR9⁻ tumor cells. As shown in new Figure 6C (and reported on page 8, lines 19-25 and page 9, lines 1-3) we found that CCR9 deficient tumor cells were efficiently lysed by T cells – irrespective of the supernatant used – while CCR9 proficient tumor cells were poorly lysed - again independent of the supernatant present. Thus, CCR9's immune modulating function is not mediated through secretion of CCL25 or other immune suppressive factors, suggesting a direct interaction with a yet unknown receptor on T cells.

In a fourth set of experiments we asked whether signaling through CCR9 in the tumor cells is required for the decreased cytotoxic activity of T cells which would favour the interpretation of an indirect effect on T cells (e.g. through upregulation of other immune modulatory surface molecules such as PDL1). We therefore treated CCR9 positive tumor cells with pertussis toxin which blocks CCR9 signaling in the tumor cells and compared the effect on T cell mediated lysis with that obtained by CCR9 knock down. The results are shown in new Figure 6D, new Supplementary Figure S4B and demonstrate that CCR9 signaling in tumor cells has no impact on T cell mediated cytotoxicity. We reported these findings in the revised results section of the manuscript (page 9, lines 3-11).

Taken together, we can conclude from these additional experiments that CCR9 exerts a direct, tumor extrinsic effect on T cells, most likely through interaction with a yet undefined immune modulatory receptor on T cells. In addition to reporting these findings in the revised results section, we discussed them in the revised discussion section (page 13, second paragraph, lines 7-10).

Query 2: The authors state on page 7 that in the Phospho-plex analysis they "observed no impact of CCR9 on TCR signaling"; however, there seem to be differences between high and low exposure to CCR9 in pLCK, and non-concentration (high vs low) impact on pCREB, pCD3e, and pERK. If the latter changes are a result solely of exposure to MCF7 cells independent of CCR9 that should be established with truly CCR9 deficient MCF7 cells (CRISPR/CAS knockout for example if a >95% siRNA knockdown is not possible) or by blockade of CCR9 with antibodies.

Response: The results regarding CCR9-dependent TCR signaling are shown in the Supplementary Figure S2. Here, we compared the response of a survivin-specific T cell clone towards CCR9 proficient or deficient MCF7 cells. As a positive control we stimulated the T cells polyclonally with PMA/Ionomycin instead of exposure to tumor cells. As polyclonal stimulation with PMA/Ionomycin should activate the T cells much stronger than MCF7 cells, we expected to see differences between the positive control and the test samples. This was indeed the case for pCREB, pCD3 and pERK. However, these factors were not found to be differentially activated after exposure to CCR9 proficient or deficient tumor cells. Regarding pLck we found a reduced activation after 5 minutes of exposure to CCR9 low tumor cells, while no differences could be seen after 1 or 20 minutes.

Taken together, these data do not strongly support the conclusion that TCR signaling is a major target of CCR9 mediated immune suppression. However, we agree that the difference in Lck activation is notable and therefore we revised the description of the respective findings in the results section as follows (page 7, second paragraph, lines 6-9): "With the exception of some degree of reduced Lck activation (which was detectable only 5 minutes after exposure to CCR9^{lo} tumor cells), we did not observe any CCR9-dependent changes in TCR signaling". As the reviewer's request to some extent may have been caused by a misunderstanding of the figure labels, we relabeled the legend in the Supplementary Figure S2.

Query 3: On page 8 the authors assert that "Indeed, CCR9 expressed on MCF7 cells selectively inhibited the secretion of T-helper-1 (Th1) cytokines such as IFN- γ , interleukin-2 (IL-2) and tumor necrosis factor-alpha (TNF- α), as well as IL-17, but increased the secretion of the immunosuppressive cytokine IL-10 (Fig. 5c)." The changes in IFN-g and IL-10, although potentially significant by T-test, do not appear biologically significant given that the "increase" in IL-10

appears to be about 2pg/mL which is biologically meaningless and outside the resolution range of the Biorad kit for IL-10. For IFN γ the "decrease" of perhaps 5-10pg/mL is at least within the actual resolution range of the Biorad kit, but is of questionable biologic significance.

Response: *We agree that the change in IL-10 secretion is minor. However, the IL-10 reduction was reproducibly significant. We used only low numbers (10.000/well) of T cells in the assay which may explain the low absolute cytokine concentrations and the supposedly low absolute differences between the test groups. In a relative scale, IL-10 was consistently reduced by 30% and IFN- γ increased by 35%. We are therefore hesitant to simply ignore these changes, but agree to emphasize on the only weak differences for these two cytokines in the revised results section as follows (page 8 lines 3-6): "CCR9 expressed on MCF7 cells significantly inhibited the secretion of the T-helper-1 (Th1) cytokines including tumor necrosis factor-alpha (TNF- α), interleukin-2 (IL-2) and (to a minor extent) of IFN- γ as well as IL-17, while the secretion of IL-10 was slightly but consistently increased (Figure 5C)".*

Query 4: Later in the same paragraph the authors continue "Accordingly, we observed a significant increase in STAT1 and slight increases in STAT2 and STAT5a/b signaling in survivin-specific T cells upon coculture with CCR9lo MCF7 cells, indicating that Th1-type immune response is impeded by tumor-specific CCR9 (Fig. 5d,e)." While the STAT1 data is significant and supports their hypothesis, the STAT5 values are not significant and therefore cannot be meaningfully invoked as inhibition of a Th1 response. The STAT3 inhibition appears as though it may be significant on the other hand but is not discussed.

Response: *Regarding STAT5 we agree with the reviewer and changed the text accordingly as follows (page 8 lines 6-9): "Accordingly, we observed a significant increase in STAT1 and STAT2 signaling in survivin-specific T cells upon coculture with CCR9lo MCF7 cells, suggesting that anti-tumor type-1 immune response is impeded by tumor-specific CCR9 (Figure 5D, E)". Regarding STAT3, there was no significant difference ($p=0.28$) between the group and this has been now indicated by "n.s." on the associated graph in Figure 5D.*

Query 5: While exposure to CCR9 hi vs lo expressing MCF7 cells produces a differential gene expression profile in T cells which hints at more effective expansion via IL-2 activation and potentially better survival via CISH, the overall transcriptional profile does not support the authors statement that "these data indicate reprogramming of the T cells to an effector phenotype upon CCR9 knockdown"

Response: *We agree with the reviewer that our statement is not sufficiently supported by the gene expression data. We now rephrased the paragraph as follows (page 9, second paragraph, lines 6-10 and page 10 lines 1-12): "Immune response-related genes such as integrin alpha-2 (ITGA2) (24), lymphotoxin alpha (LTA) (25), interleukin 2 receptor alpha (IL2RA) (26), cytokine inducible SH2-containing protein (CISH) (27) were upregulated; whereas genes that inhibit T cell maturation and effector function such as ephrin-A1 (EFNA1) (28), Kruppel-like factor 4 (KLF4) (29), inhibitor of DNA binding-1 (ID1) (30), transducer of ERBB2, 1 (TOB1) (31) were downregulated in T cells encountering CCR9lo tumor cells, which was found to be in accordance with the observed increase in cytotoxicity as shown before. Gene-annotation/ontology (GO) analysis of the top up-regulated genes revealed an enrichment of genes involved in positive regulation of immune response, while genes involved in lymphocyte maturation and apoptosis were enriched in the list of downregulated genes (Supplementary Figure S6A). We next wondered if these gene signatures associated with re-activated T cells upon tumor-specific CCR9 knockdown overlap with gene signatures generally associated with an activated T cell population. Using a publically available gene expression study comparing unstimulated CD8+ T cells to CD3/CD28 antibody and IL-2 activated T cells (32), we indeed identified overlapping gene signatures in both these studies (Supplementary Figure S6B), suggesting that CCR9 knockdown on tumor cells favors better survival, proliferation and activation of the encountering T cells".*

Reviewer #2:

We thank the reviewer for careful consideration of our manuscript and invaluable suggestions for improvement.

Remarks: This paper describes a novel screening methodology to identify cell surface molecules on cancer cells involved in down-regulation of T cell responses. The screening method is novel, simple and reliable. Using this method, the authors have identified several novel molecules involved T cell functional down-regulation, notably of cytotoxic T cell function. The molecule CCR9 has not previously been implicated as a cancer cell surface molecule involved in such suppression. Its role was investigated in detail, and found to be based on direct down regulation of T cell function. Moreover CCR9 was found to exert the same negative role on other cancer cell lines. This work can lead to novel approaches to intervention to promote cancer immunotherapy involving effector T cells.

Query 1: The one obvious question is how important CCR9 is in the overall inhibition of T cell function in vivo. The authors can show this by blocking CCR9 with antibodies in vivo or by showing that CCR9 KO cells are better eradicated in vivo than the parental CCR9+ tumor cells. This would require demonstration that CCR9 fulfills the same role on murine cancer cells and T cells.

Response: *We thank the reviewer for this important request. We have meanwhile conducted tumor rejection experiments in vivo using CCR9 proficient and –deficient variants of the melanoma cell line M579 which we generated through stable shRNA expression using lentiviral transduction. We now show in the new Figure 7A-D that CCR9 wt tumors cannot be controlled by adoptively transferred tumor-specific TIL, while tumor growth of the CCR9 deficient counterparts is efficiently inhibited after TIL transfer.*

Thus, CCR9 expression by tumor cells efficiently inhibits T cell based anti-tumor immunotherapy in vivo. These findings are reported in the revised results and discussion section (page 10, second paragraph; page 11, lines 1-5 and page 13, third paragraph, line 1-3).

Query 2: Potential side effects of CCR9 blocking should also be investigated and discussed.

Response: *Based on available literature on a CCR9 knockout mouse model we have now added a comment on potential side effects of CCR9 knock down in the revised discussion section as follows (page 13, second paragraph, lines 5-9 and page 14, lines 1-4): “Further development of CCR9 as an immunotherapeutic target for cancer treatment would however require extensive toxicological analysis. Previous study with CCR9 knockout mouse model has reported no adverse side effects of CCR9 gene ablation, besides the reduction in the $\gamma\delta$ IELs which are characterized by their CCR9+ phenotype (42). However, given that CCR9 is important for the homing of effector T cells to the gut, systemic targeting of CCR9 would be a contraindication for gut-associated tumors. On the other hand, CCR9 was found to be expressed on only a small minority of survivin-specific T cells (~ 6%) and was undetectable on peripheral blood T cells from healthy donors (data not shown). Its expression in other immune cell subsets and that on healthy tissue remains to be examined”.*